# Natural Cyclopeptides as Anticancer Agents in the Last 20 Years

**DOI:** 10.3390/ijms22083973

**Published:** 2021-04-12

**Authors:** Jia-Nan Zhang, Yi-Xuan Xia, Hong-Jie Zhang

**Affiliations:** Teaching and Research Division, School of Chinese Medicine, Hong Kong Baptist University, Kowloon, Hong Kong SAR, China; 14253674@life.hkbu.edu.hk (J.-N.Z.); xiayixuan@hkbu.edu.hk (Y.-X.X.)

**Keywords:** cyclopeptides, anticancer, natural products

## Abstract

Cyclopeptides or cyclic peptides are polypeptides formed by ring closing of terminal amino acids. A large number of natural cyclopeptides have been reported to be highly effective against different cancer cells, some of which are renowned for their clinical uses. Compared to linear peptides, cyclopeptides have absolute advantages of structural rigidity, biochemical stability, binding affinity as well as membrane permeability, which contribute greatly to their anticancer potency. Therefore, the discovery and development of natural cyclopeptides as anticancer agents remains attractive to academic researchers and pharmaceutical companies. Herein, we provide an overview of anticancer cyclopeptides that were discovered in the past 20 years. The present review mainly focuses on the anticancer efficacies, mechanisms of action and chemical structures of cyclopeptides with natural origins. Additionally, studies of the structure–activity relationship, total synthetic strategies as well as bioactivities of natural cyclopeptides are also included in this article. In conclusion, due to their characteristic structural features, natural cyclopeptides have great potential to be developed as anticancer agents. Indeed, they can also serve as excellent scaffolds for the synthesis of novel derivatives for combating cancerous pathologies.

## 1. Introduction

Cancer has become an enormous burden in society and a leading cause of death in the world. Apart from aging, the risk factors of modern daily life such as smoking, unhealthy diet and mental stress also play significant roles in the growing incidence of cancer [1]. According to the World Health Organization (WHO), nearly 10 million of people were killed by cancer in 2020. New cancer cases reached to 18 million in 2018, and are expected to grow to up to 27.5 million in 2040 [2,3]. Conventional therapeutic approaches for the treatment of cancer include chemotherapy, radiotherapy and surgical resection. In recent years, advancement in cancer treatment has been applied with some novel therapies and choices for patients to improve their survival chances, such as immunotherapy, targeted therapy, hormone therapy, stem cell transplant and precision medicine [4]. However, low specificity to cancer cells, high toxicities to normal tissues as well as drug resistance are often the most problematic concerns [5,6,7]. Therefore, studies on the development of new drugs and therapies for cancer with higher efficacy are urgent and crucial.

From ancient times, taking Chinese Medicine as an example, people have been familiar with medicinal properties of natural resources and the benefits of using them to treat diseases [8]. Hence, due to their extraordinary chemical diversity with different origins, natural products are of great potential to be developed as new drugs for cancer prevention and treatment [9]. Natural cyclopeptides, due to their special chemical properties and therapeutic potency have attracted great attention from both academic researchers and pharmaceutical companies in recent years.

Cyclopeptides, also known as cyclic peptides, are polypeptides formed from amino acids arranged in a cyclic ring structure [10]. Most natural cyclopeptides are found to be assembled with 4–10 amino acids. However, some natural cyclopeptides can contain dozens of amino acids [11,12,13,14]. In the past 20 years, hundreds of ribosomally synthesized cyclopeptides have been isolated from a variety of living creatures with various biological activities. In fact, there are two biosynthetic mechanisms of cyclopeptides—ribosomal and non-ribosomal pathways. The ribosomal pathway merely requires the 21 basic proteinogenic amino acids for the synthesis of cyclopeptides, whereas the non-ribosomal pathway depends on non-ribosomal peptide synthetases (NRPSs) in incorporating functional groups, such as hydroxyl and N-methyl groups, to the existing peptides [15,16]. A large number of cyclopeptides have been isolated and demonstrated to deliver a wide spectrum of bioactivities. Very often, they can be served as antibiotics (e.g., microcin 25 from *Escherichia coli*) [17], protease inhibitors (e.g., SFTI-1 from *Helianthus annuus*) [18], boosting agents for the innate immune system (e.g., θ-defensins from *Macaca mulatta*) [19], anticancer agents (e.g., chlamydocin from *Diheterospora chlamydosporia*) [20] and so on. According to previous reports, cyclization is an important process for peptide stabilization [21]. Compared with linear peptides, cyclic peptides may have a better biological potency due to the rigidity of their conformation [22]. Moreover, the rigidity of cyclopeptides may also increase the binding affinity to their target molecules, allowing enhanced receptor selectivity as rigid structures often decrease the interaction entropy to a negative Gibbs free energy. Furthermore, cyclopeptides are resistant to exopeptidases because they contain no amino/carbonyl termini in their skeletons, so that they exert more stable binding capability than linear peptides do. Owing to the absence of the amino/carbonyl termini, the less flexible backbone of cyclopeptides even enhances their resistance to endopeptidases, and therefore, cyclopeptides are considered as useful biochemical tools with high specificity [23]. In particular, when serving as an anticancer agent, the cyclic structure of the peptide eliminates the effect of charged termini and thus increases membrane permeability [24]. As a result, cyclopeptides can penetrate to the tumors without much hurdle while exhibiting their anti-proliferative effect [25].

Overall, the structural rigidity, biochemical stability, binding affinity and membrane permeability are the predominant advantages of cyclopeptides in comparison of linear peptides and other chemicals for therapeutic purposes [10]. Owing to these exceptional characteristics, natural cyclopeptides are considered as promising lead compounds for the development of novel anticancer drugs. With the efforts from researchers around the world, a great number of cyclopeptides with anticancer effects have been identified from various living entities. In this review, we summarize the recent development of cyclopeptides with anticancer effects according to their origins.

## 2. Marine-Derived Anticancer Cyclopeptides

Marine organisms are the vast source of many chemical products because of the diversity of species and their abundance [26]. In fact, the marine environment though somehow mysterious, promotes the fruitful development of a great number of natural products, which may possess novel structures, chemical properties and biological effects [27]. The marine-derived cyclopeptides have received great attention due to their high potency in anti-microbial, anti-inflammatory and anticancer activities [28,29,30,31]. This section mainly discusses the discovery of marine-derived cyclopeptides with anticancer properties in the last 20 years.

### 2.1. Anticancer Cyclopeptides Derived from Sponge

As sessile marine filter feeders, sponges possess decent abilities to defense against foreign invaders such as viruses, bacteria and fungi [32]. Marine sponges belong to the phylum Porifera, which is the source of many bioactive marine natural products [33]. It is estimated that about 5000 different new compounds have been discovered from sponges each year [32]. Among them, cyclopeptides are an important portion of the sponge-derived compounds that are accompanied with broad biological effects including anticancer activities [34]. The structures of these sponge-derived natural cyclopeptides compounds are shown in Figure 1 and their in vitro activities against different cancer cells are listed in Table 1.

Azumamides A (**1**) and E (**2**) were two cyclotetrapeptides isolated from the marine sponge *Mycale izuensis*. These two compounds have been demonstrated to exhibit potent inhibitory activities against histone deacetylase (HDAC) enzymes [35]. It is widely acknowledged that overexpression of HDACs may lead to the occurrence of heterochromatin and thus different types of cancers [36]. In consequence, HDAC inhibitors are suggested with a role in cancer growth inhibition, and their anticancer mechanisms are mainly associated with their induction of apoptotic events in cancer cells [37]. According to a previous study, azumamide A showed anti-proliferative effect on K562 (human bone marrow chronic myelogenous leukemia) cells with IC_50_ values of 0.045 μM, and cytostatic effect on WiDr (human colon cancer) and K562 cells with IC_50_ values of 5.8 and 4.5 μM, respectively [35]. Furthermore, azumamides A and E obtained from total synthesis were demonstrated to exert significant zinc-binding affinity to HDACs, which may contribute to the rational design for synthesizing potent anticancer agents [38,39].

Microsclerodermin A (**3**), a cyclic hexapeptide, first isolated from the lithistid sponge *Microscleroderma herdmani*, was found to be a potent antifungal agent [40,41]. This compound was also reported to have anti-proliferative activities against lung, leukemia and pancreatic cancer cells. In human pancreatic adenocarcinoma cells (AsPC-1), the level of phosphorylated NFκB was significantly reduced by the incubation with microsclerodermin A (IC_50_ = 2.3 µM) whereas apoptosis was strongly induced. In the human pancreatic adenocarcinoma cells BxPC-3 (IC_50_ = 0.8 µM) and human pancreatic epithelioid carcinoma PANC-1 cells (IC_50_ = 4.0 µM), microsclerodermin A exhibited remarkable inhibitions at low IC_50_ values. In human umbilical vein endothelial cells (HUVECs), this cyclopeptide was shown to have no effect of angiogenesis [42]. The above biological investigations demonstrated that microsclerodermin A is a good anticancer lead compound for further development.

Leucamide A (**4**) is a cytotoxic cyclic heptapeptide with potent cytotoxicity, which was first purified from the Australian marine sponge *Leucetta microraphis*. In the study of Kehraus and co-workers, this peptide was found to exhibit inhibitory activities against human mucin-producing gastric cells (HM02), and human liver cancer cells (HepG2 and Huh7) with GI_50_ values of 8.5, 9.7 and 8.3 µM, respectively [43]. In the consecutive year, the total synthesis of leucamide A was accomplished, and provided a viable approach to the sufficient supply of this compound for further comprehensive investigations [44].

Afifi et al. isolated a new cyclic heptapeptide, named carteritin A (**5**), from the marine sponge *Stylissa carteri*. Carteritin A exhibited cytotoxicities against human colorectal (HCT116), mouse macrophage (RAW264) and human cervical (HeLa) cancer cells with IC_50_ values of 1.3, 1.5 and 0.7 µM, respectively [45].

Stylissatin B (**6**), a novel cytotoxic cycloheptapeptide, was obtained from the sponge *Stylissa massa*. This compound displayed inhibitory effects on HCT116, HepG2, gastric (BGC823), lung (NCI-H1650), ovarian (A2780) and breast (MCF7) human cancer cells with IC_50_ values ranging 2.3 to 10.6 µM [46]. These results implicated that significant structural modification is needed to improve the moderate activity of Stylissatin B.

Derived from the sponge *Theonella cupola*, cupolamide A (**7**) was found to be a biologically active cyclic heptapeptide. This compound was cytotoxic against mouse lymphoma cells (P388) with an IC_50_ value of 7.5 µM but no cytotoxic against thrombin. Moreover, the resemblance of cupolamide A to γ-aminobutyric acid (GABA) unveils its potential effect on the central nervous system, which requires further investigation [47].

Geodiamolide H (**8**), a cyclic depsipeptide isolated from the Brazilian sponge *Geodia corticostylifera*, was firstly identified as a neurotoxic and hemolytic agent [48]. This compound showed a potent anti-proliferative activity against the human ductal carcinoma T47D (EC_50_ = 38.36 nM) and MCF7 (EC_50_ = 89.96 nM) cells via altering the cytoskeleton of cancer cells by actin depolymerization. However, no such deleterious effect was observed in normal cell lines even when higher concentrations was used (e.g., 136 nM) [49]. Subsequently, the results of another study further demonstrated that geodiamolide H exhibited significant inhibitory effects on the migration and invasion of breast cancer cells (Hs578T) that were poorly differentiated and highly aggressive [50]. These studies revealed geodiamolide H as a potential lead compound for further development of therapeutic use.

Theopapuamide (**9**), isolated from the lithistid sponge *Theonella swinhoei*, was also a new cytotoxic depsipeptide with a cyclic ring structure. This cyclopeptide exhibited significant cytotoxicity against human leukemia (CEM-TART) (EC_50_ = 0.5 µM) and HCT116 (EC_50_ = 0.9 µM) cells. When compared with other depsipeptide with HIV-inhibitory activities, the *β*-methoxytyrosine residue of this compound appears to be of great significance for its anti-HIV biological effect [51].

A group of structurally similar cyclodepsipeptides named homophymines A-E (**10**–**14**) and A1-E1 (**15**–**19**) isolated from the sponge *Homophymia* sp., were found to be potent anti-proliferative agents against a wide panel of cancer cell lines with IC_50_ values in the range of 2–100 nM. Among these cell lines, human prostate (PC3) and ovarian (OV3) human cancer cell lines were the most sensitive to this group of cyclopeptides. Furthermore, the overexpression of P-glycoprotein (P-gp) proteins did not affect the intracellular concentrations of homophymines [52]. Apart from cytotoxic activities, homophymine A also exhibited cytoprotective activity against HIV-1 infection (IC_50_ = 75 nM) [53]. In the family of homophymins, their general chemical structures are complicated but structural differences among them are subtle. Because of their significant anticancer potential, further structure activity relationship (SAR) assessment, bioactivity evaluation and mechanism of action studies should be performed.

Jaspamide is an early-discovered cyclic depsipeptide from the sponge *Jaspis splendens*, which delivered pronounced inhibitory activities against breast and prostate cancer cells [31]. Pipestelide A (**20**), a cyclodepsipeptide derived from the marine sponge *Pipestela candelabra*, was biosynthetically related to jaspamide. When subjected to cytotoxicity assay, this compound showed cytotoxicity against the human nasopharyngeal epidermoid carcinoma (KB) cells with an IC_50_ value of 0.1 µM, which was less potent than jaspamide [54].

Three cytotoxic cyclodepsipeptides, neamphamides B-D (**21**–**23**), were purified from the Australian sponge *Neamphius huxleyi*. These three compounds were cytotoxic to human lung (A549), cervical (HeLa) and prostate (LNCaP and PC3) cancer cells with IC_50_ values varying from 91 to 230 nM. Nevertheless, at sub-nanomolar concentrations (100 nM for 48 h and 72 h), neaphamide D was observed to induce proliferation in human lung cancer A549 cells. Therefore, these neamphamindes should be discreetly utilized [55]. Apart from the cytotoxic effects, neaphamide D also showed a great antimicrobial potency against the growth of *Mycobacterium smegmatis* and *M. bovis* BCG [56].

Collected in Solomon Islands, the sponge *Asteropus* sp. yielded two new derivatives of callipeltin A, which were named cyclic depsipeptides callipeltins N (**24**) and O (**25**). Their cytotoxicities against human melanoma (A2058), colorectal (HT29), and breast (MCF7) cancer cells as well as non-malignant MRC-5 fibroblast cells were accompanied with IC_50_ values ranging from 0.16 to 0.21 µM for callipeltins N and ranging from 0.48 to 2.08 µM for callipeltins O. The slight difference of cytotoxicities between malignant cancer cells and non-malignant cells indicated that these two compounds can provide minimal specificity.Structure-wise, the biological potencies of the two compounds was highly related to the positions of the methylGln group [57].

Pipecolidepsins A (**26**) and B (**27**), two head-to-side-chain cyclodepsipeptides with significant anticancer potential, were isolated from the sponge *Homophymia lamellosa*. Both compounds were tested for their cytotoxic effects against A549, HT29 and human breast cancer (MDA-MB-231) cells. The results demonstrated that pipecolidepsin B (GI_50_ = 0.04, 0.01 and 0.02 µM, respectively) was more cytotoxic than pipecolidepsin A (GI_50_ = 0.6, 1.12 and 0.7 µM, respectively), though pipecolidepsin B contains only one more hydroxy group. The SAR assessment further revealed that the replacement of residue at C-3 was an important step for gaining higher hydrophilic properties in pipecolidepsin B [58], whereas the γ-amino acid in pipecolidepsin A was the major functional group for its cytotoxicity [59], While exerting cytotoxic activity, pipecolidepsin A was shown to induce intense membrane damage and necrotic cell death [60]. Due to the early discovery of pipecolidepsin A, a total solid-phase synthesis of this compound had been previously accomplished [61]. Considering the great anticancer potential of pipecolidepsin B, the further understanding of its biological property, total synthetic strategy as well as other pharmacological actions is of high research interest.

Halipeptin D (**28**) was a cyclic depsipeptide isolated from the sponge *Leiosella* cf. *arenifibrosa* by Faulkner and Manam [62]. This natural product was reported to exhibit significant cytotoxicity against HCT116 cells (IC_50_ = 7 nM) and BMS ODCP (oncology diverse cell panel) with an average IC_50_ value of 420 nM. However, the synthetic halipeptin D and other halipeptins (e.g., halipeptins A-C) did not show potent cytotoxicities. Based on the distinct results, it is reasonable to assume that the previous naturally derived compound might have been contaminated with other cytotoxic agents during harvest [63]. The discrepancy of cytotoxicity between the synthetic and naturally derived compounds is not surprising as such happens somehow. As a result, biological properties of halipeptin D remained uncertain and required further investigations.

The cyclic peptide reniochalistatin E (**29**) was isolated from the marine sponge *Reniochalina stalagmitis* by Zhan and co-workers. Differed from other congeners (reniochalistatins A-D) of the same origin, this compound showed in vitro cytotoxicity against RPMI-8226 (human myeloma) and MGC-803 (human gastric carcinoma) cells with IC_50_ values of 4.9 and 9.7 μM, respectively [64]. The subsequent total synthesis of reniochalistatin E provided some inspiring methods for the synthetic strategies of cyclopeptides with similar structures [65].

From the freeze-dried Indonesian sponge *Callyspngia aerizusa*, a cyclic peptide called callyaerin G (**30**) was purified. Biological studies showed that this compound was cytotoxic to mouse lymphoma cells (L5178Y) and HeLa cells with ED_50_ values of 0.41 and 4.17 μM, respectively [66]. Callyaerins E (**31**) and H (**32**), were purified using another bioassay-guided fractionation of the same sponge *Callyspngia aerizusa*. Callyaerins E and H exhibited strong anti-proliferative activities against L5178Y cells with ED_50_ values of 0.39 and 0.48 µM, respectively. Similar to the three compounds above, other proline-rich callyaerins yielded from the same sponge also showed high antimicrobial activities [67]. Accordingly, the biological significance of proline residue in cyclic peptides becomes a trendy research topic. The mechanism of callyaerins in biological actions definitely deserves further investigation.

Obtained from the lithistid sponge *Scleritoderma nodosum*, scleritodermin A (**33**) was a cyclic peptide with anticancer potential. By means of cell viability assay, this compound was demonstrated to exert cytotoxicity in various cancer cell lines including HCT116, HCT116/VM46 (human vinblastin-resistant colon cancer cell), A2780 and SKBR3 (human breast cancer cells) with IC_50_ values of 1.9, 5.6, 0.94 and 0.67 µM, respectively. Regarding its molecular mechanism, scleritodermin A induced G2/M phase arrest and inhibited tubulin polymerization [68]. By using different moieties and fragments, the total synthesis of scleritodermin A had been accomplished [69,70], and consequently provided a sufficient supply of this compound for further comprehensive studies of its other biological properties and the synthesis of analogs or derivatives.

Calyxamides A (**34**) and B (**35**), two novel cyclic peptides with bioactivities, were first isolated from the marine sponge *Discodermia calyx* in Japan. When incubated with leukemia P388 cells, both compounds yielded moderate cytotoxicities with IC_50_ values of 3.9 and 0.9 µM, respectively. Subsequently, the analysis of the 16S rDNA sequence indicated that the biosynthesis of calyxamides A and B were possibly related to Candidatus *Entotheonella* sp. inhabiting in the *Theonella* genus [71]. Hence, further biological studies of these compounds would be relied on the bulky isolation from their natural source.

A group of compounds named keramamides were yielded from the Okinawa marine sponge *Theonella* sp., which possess a variety of bioactivities. Keramamide E (**36**) was reported to display inhibitory activities against mouse lymphocytic leukemia (L1210) and KB cells with IC_50_ values of 1.42 and 1.38 µM, respectively [72]. Subsequent explorations on another *Theonella* sponge led to the purification of keramamides K (**37**) and L (**38**). Keramamide K contains a (1-Me)Trp residue, which is rarely seen in any naturally derived compound whilst keramamide L is a natural cyclopeptide possessing a MeCtrp residue. These two special cyclopeptides exhibited strong cytotoxicity against L1210 (IC_50_ = 0.77 and 0.5 µM, respectively) and KB (IC_50_ = 0.45 and 0.97 µM, respectively) cells [73]. Keramamides M (**39**) and N (**40**) were two other congeners from the *Theonella* sponge that were discovered by the same research team. Both cyclic peptides contained a sulfate ester, which is also a rare residue to be found in compounds derived from marine sponge. Keramamides M and N showed moderate cytotoxicities against L1210 (IC_50_ = 2.02 and 2.33 µM, respectively) and KB (IC_50_ = 5.05 and 6.23 µM, respectively) cells [74]. The total synthesis of different keramamides had been completed by the same research group [75]. From the results of structural elucidation, some intriguing constituents of these cyclic peptides exhibited anticancer potentials against some other cancer cell lines. In order to highlight the aspect of their anticancer effect, more types of cancer cell lines and mechanism of their biological actions should be exploited.

The marine sponge *Axinella carteri* collected near Vanuatu islands yielded two proline-containing cyclopeptides, which were called axinellins A (**41**) and B (**42**). Both compounds were found to display moderate cytotoxicity against non-small lung cancer cells (NSLC-N6) with IC_50_ values of 16.7 and 29.8 µM, respectively [76]. The first synthesis of axinellin A was achieved shortly after its discovery. However, the synthetic compound did not show cytotoxicity as the naturally occurring peptide did [77]. Therefore, further investigation should be performed for clarifying such discrepancy on cytotoxicities between the synthetic and the natural axinellin A.

The proline-rich cyclopeptide stylopeptide 2 (**43**) was isolated from the marine sponge *Stylotella* sp. in Papua New Guinea. According to the study of Brennan et al., this compound inhibited 23% growth of BT-549 and 44% growth of Hs578T human breast cancer cells in the National Cancer Institute one-dose (10^−5^ M) 60-cell-line assay [78]. However, further dose-dependent cell viability assays should be carried out for a more accurate evaluation of its cytotoxicity and anticancer potential.

Stylissamide X (**44**) was a cyclic proline-rich octapeptide discovered in the Indonesian marine sponge of *Stylissa* species. Although only little inhibitory activity was observed in the viability assay of HeLa cells, stylissamide X induced significant anti-migration effects on the same cell line in the wound-healing assay [79]. As a new member of the stylissamides family, the total synthetic approach of it became extremely important for securing a good supply of materials for further comprehensive chemical and biological studies [80]. More systemic cell viability assays and detailed evaluation of its anti-migration properties would be of similar importance for the mode of action studies of this type of compounds.

Aciculitins A-C (**45**–**47**) were three bicyclic peptides extracted from the lithistid sponge *Aciculites orientalis* with an unusual histidino-tyrosin bridge. They were also the first glycopeptidolipids obtained from a marine origin. In regards to their biological activities, aciculitins A-C were cytotoxic against HCT116 (IC_50_ = 0.37 µM) and inhibitory to the growth of bacterium *Candida albicans*. Comparing with other compounds of the same origin, aciculitins A-C possess a histidine residue which is responsible for their predominant bioactivities [81].

Nazumazoles A-C (**48**–**50**) were three novel bicyclic pentapeptides isolated from a mixture derived from the marine sponge *Theonella swinhoei*. The mixture of nazumazoles A-C showed cytotoxic activities against P388 cells with an IC_50_ value of 0.83 µM. The reduction of either the ketone or thiol group in the nazumazoles led to a significant decrease of the cytotoxicity of the mixture [82]. Based on the results above, separation of individual nazumazoles A-C is undoubtedly needed to validate the anticancer potential of this type of compounds.

Theonellamide G (**51**) was a novel bicyclic glycopeptide isolated from the Red Sea Sponge *Theonella swinhoei*. When compared the positive anticancer control etoposide (2.0 µM), theonellamide G merely yielded a moderate cytotoxicity against HCT116 with an IC_50_ value of 6.0 µM. However, this compound showed a potent antifungal activity against the wild and amphotericin B-resistant strains of *Candida albicans* [83]. These results added some fresh insights into the diverse biological activities of this natural compound, which may inspire further studies on the pharmacological actions and SAR for this class of compounds.

Koshikamide B (**52**) was a 17-residue cyclic peptide lactone purified from a marine sponge of the genus *Theonella*. This compound was the first natural product containing the constitution of *N*^δ^-carbamoylasparagine. Importantly, the 2-(3-amino-2-hydroxy-5-oxopyrrolidin-2-yl) propionic acid (AHPP) residue of koshikamide B was a unique moiety in peptide lactones. Koshikamide B exhibited cytotoxicity in HCT116 and P388 cancer cells with IC_50_ values of 3.62 and 0.22 µM, respectively [84].

### 2.2. Anticancer Cyclopeptides Derived from Ascidians/Tunicates

Ascidians are also known as tunicates. They are found in high-current fields, and firmly fixed to rocks. About 3000 species of ascidians have been reported so far [85]. As more and more compounds were isolated from the ascidians, the US National Cancer Institute (NCI) estimated that about 1% of the marine natural products (MNPs) showed anticancer activities [86]. This section focuses on the ascidians derived cyclopeptides with promising anticancer potential. The structures of these compounds are present in Figure 2 and their activities are listed in Table 1.

Mollamide (**53**) was a cyclopeptide isolated from the ascidian *Didemnum molle*. This compound displayed a moderate cytotoxicity against several cancer cell lines, with IC_50_ values of 1.24 μM against P388 cells, and 3.1 μM against A549, HT29 and monkey kidney CV1 cells. The anti-proliferative property of mollamide was plausibly associated with its ability to inhibit RNA synthesis in the cancer cells [87,88]. Another research study on the same Indonesian tunicate *Didemnum molle* led to the isolation of a novel congener, which was a cyclic hexapeptide called mollamide B (**54**). Mollamide B (100 μM) displayed significant growth inhibition against human lung carcinoma (H460), MCF7 and glioblastoma (SF-268) cells by 29%, 44% and 42%, respectively. Other than its anticancer activities, mollamide B also exhibited notable anti-HIV (HIV-1, EC_50_ = 48.7 μM) and anti-malarial (*Plasmodium falciparum*, IC_50_ = 2.87 μM) properties [89]. For the future perspective of its anticancer potential, the specificity of mollamide B acting against different cancer cell lines should be investigated.

Trunkamide A (**55**) is an analog of mollamide with a similar cyclopeptide structure that is naturally produced by ascidians of *Lissoclinum* sp. Compared to mollamide, trunkamide A was reported to have a more promising antitumor activity [90]. Due to its significant in vitro cytotoxicity against several human-derived cancer cell lines including A549, P388, HT29 and MEL28 (human melanoma) with IC_50_ values ranging from 0.6 to 1.19 μM, trunkamide A had been evaluated for its antitumor efficacy in several preclinical trials [91].

Fu et al. purified a novel cyclic peptide called prepatellamide A (**56**), from *Lissoclinum patella*. Although the structure of this peptide had been determined, its biological properties were incompletely understood. The IC_50_ value of prepatallamide A against P388 cells was 6.57 µM [92]. Other than this piece of result, no further information on prepatallamide A could be obtained. Therefore, the bioactivity studies of this compound are desperately needed.

Vitilevuamide (**57**), another ascidian derivative with a bicyclic structure, was found in two species of ascidians, *Didemnum cuculiferum* and *Polysyncranton lithostrotum*. It displayed potent cytotoxicity against HCT116 cells, with an IC_50_ value of 6.24 nM. In A549, SKMEL-5 (human melanoma) and A498 (human kidney carcinoma) cells, vitilevuamide exhibited a less potent anti-proliferative effect as its IC_50_ values were determined to be 0.12, 0.31 and 3.12 μM, respectively. In addition, the compound showed a percentage increase in the mean lifespan (%ILS) of 70 at the concentration of 30 μg/mL in a P388-xenograft in vivo experiment in mice, which indicated its significant anticancer potential [93]. Regarding its anticancer mechanisms, vitilevuamide inhibited tubulin polymerization and induced cell cycle arrest at the G2/M phase [94]. As the supply of the naturally derived vitilevuamide is limited, further evaluations on this compound are largely hindered. Due to its complex structure, identification of its interactive site(s) with tubulin and methods of chemical synthesis are in slow progress.

Tamandarin A (**58**) was first isolated from a Brazilian marine ascidian of the family Didemnidae. This compound was cytotoxic to three cancer cell lines, BX-PC3 (pancreatic carcinoma), DU145 (prostate carcinoma) and UM-SCC-10B (laryngeal squamous cell carcinoma) with IC_50_ values of 1.69, 1.29 and 0.94 nM, respectively. Didemnin B, another renowned tunicate-derived cyclodepsipeptide, had been promoted into clinical trials several years ago owing to its significant anticancer potential [95]. With slightly lower IC_50_ values than didemnin B, tamandarin A can also be a good anticancer candidate [96,97]. Although its mechanism of action remains unclear, the structural similarity between tamandarin A and didemnin B may represent their similar mechanisms of action in cancer cells, which warrants it for further biological investigations.

Rudi’s research group found two groups of cyclic hexapeptides, comoramides A and B (**59**–**60**) and mayotamides A and B (**61**–**62**), which were isolated from the ascidian *Didemnum molle* collected at different spots. Cytotoxicity screening revealed that these two groups of compounds showed cytotoxic activities against A549, HT29 and MEL28 cell lines with IC_50_ values ranging from 7.22 to 14.97 µM [98].

Patellin 6 (**63**) was a cytotoxic cyclic hexapeptide isolated from the colonial ascidian *Lissoclinum patella*. Although patellins 1-5 exhibited no cytotoxicity, patellin 6 was highly cytotoxic to P388, A549, HT29 and CV1 cells with the IC_50_ values around 2.08 µM. In addition, patellin 6 inhibited the activity of topoisomerase II with an IC_50_ value of 2.6 µM [99].

Cycloforskamide (**64**) was a macrocyclic dodecapeptide isolated from the sea slug *Pleurobranchus forskalii* collected in Japan. This compound showed cytotoxic effect against P388 cells with an IC_50_ value of 5.8 µM. In addition, from an ecological perspective, cycloforskamide is presumed to chelate toxic metals and plays a detoxification role [100].

### 2.3. Anticancer Cyclopeptides Derived from Mollusks

As the largest marine phylum and the secondary largest phylum of invertebral animals, mollusks possess highly diverse anatomical structures while having various behavior and habitats [101]. Surprisingly, a great amount of enzymes, polysaccharides, lipids and peptides with broad therapeutic uses have been discovered from mollusks. No wonder mollusks are considered a great resource for the discovery of bioactive compounds [102]. Many previous studies reported that mollusk-derived compounds showed significant anticancer properties, and many of them are indeed cyclopeptides. This section lists a wide range of mollusk-derived cyclopeptides with decent anticancer activities. The structures of these compounds are shown in Figure 3 and their activities are listed in Table 1.

Keenamide A (**65**) is a cyclic hexapeptide isolated from *Pleurobranchus forskali*. This compound elicited anti-proliferative activities against tumor cell lines A549, P388, MEL20 (uveal melanoma) and HT29 with IC_50_ values ranging from 4.03 to 8.05 μM; however, its molecular mechanisms remained elusive [103].

Aurilide (**66**) is a 26-membered cyclic peptide extracted from the internal organs of sea hare *Dolabella auricularia*. It exhibited significant cytotoxicity against HeLa S_3_ tumor cells with an IC_50_ value of 0.013 µM. In the NCI-60 assay, this compound showed a strong inhibitory activity against the ovarian, renal as well as prostate cancer cells. However, no significant antitumor effect was observed in the subsequent in vivo experiments in xenograft mice. In regards to its antitumor mechanism, aurilide promoted microtubule stabilization rather than inducing a direct tubulin interaction [104]. Sato et al. found that aurilide could selectively bind to prohibitin-1 (PHB1) in mitochondria and activate proteolytic processing of OPA1, resulting in mitochondrial apoptosis [105]. However, further studies are needed to clarify the obvious diverging results obtained from the in vitro and in vivo tests of aurilide. In addition, detailed studies of mechanism and SAR are also needed to develop this type of cyclopeptides as druggable anticancer agents.

Dolastatin 16 (**67**) is a cyclodepsipeptide originated from the sea hare *Dolabella auricularia* in Papua New Guinea. When tested against NCI cancer cell lines, this compound exhibited strong inhibitory activities against H460 (GI_50_ = 1.09 nM), colon adenocarcinoma KM20L2 (GI_50_ = 1.37 nM), glioblastoma SF-295 (GI_50_ = 5.92 nM) and SKMEL-5 (GI_50_ = 3.75 nM) cells. Moreover, dolastatin 16 also showed comparable anti-proliferative effects against five human leukemia cell lines [106]. Although studies on the molecular mechanisms or pharmacological actions of dolastatin 16 are rather limited, this highly potent bioactive compound is worthy for further development into an anticancer agent.

Doliculide (**68**), a potent cytotoxic cyclodepsipeptide, was previously isolated from the Japanese sea hare *Dolabella auricularia*. This compound showed very potent cytotoxicity against HeLaS_3_ cells with an IC_50_ value of 1.62 nM [107]. It is worth noting that the first total synthesis of doliculide was accomplished soon after its isolation [108]. Actin is an abundant protein in many eukaryotic cells, and it is also an important target in many cancer studies. In order to target actin polymerization, Matcha et al. developed an efficient strategy for synthesizing (-)-doliculide, which was proven with great actin binding ability [109]. When applied to breast cancer cell lines MCF7 and MDA-MB-231, the synthetic doliculide showed a decent anti-proliferative activity with IC_50_ values of 93 and 77 nM, respectively. Induction of apoptotic events and migratory impairment were also observed in both cell lines when incubated with doliculide [110]. Collectively, the above experimental results unveiled the high anticancer potential of doliculide.

Kulokekahilide-1 (**69**) was a cyclic depsipeptide purified from the cephalaspidean mollusk *Philinopsis speciosa*. This compound showed cytotoxic activity against P388 cells with an IC_50_ value of 2.2 µM [111]. Kulokekahilide-2 (**70**), a much more potent cytotoxic cyclic depsipeptide than **69** was isolated from the same origin two years later. The IC_50_ values of kulokekahilide-2 in P388, human ovarian cancer (SK-OV-3), human breast cancer (MDA-MB-435) and rat myoblast (A-10) cells were determined to be 4.2, 7.5, 14.6 and 59.1 nM, respectively [112]. Due to the potent anticancer potential, kulokekahilide-2 and its derivatives were totally synthesized in subsequent studies. According to the SAR assessments and cytotoxicity tests performed by Umehara et al., the IC_50_ values of the synthesized kulokekahilide-2 against A549, K562 and MCF7 cells were found to be 0.0021 nM, 0.0031 nM and 0.22 nM. These low IC_50_ values indicated that this compound was of great anticancer potential. Interestingly, the cyclic structure of this peptide accompanied by the chirality at the 21 position largely contributes to its super potent cytotoxicity [113]. According to the NCI COMPARE analysis the mechanisms of action of kulokekahilide-2 appeared rather different from those of the conventional anticancer agents such as aurilide (**66**), palauamide (**182**) and lagunamide A (**152**) [114]. Nevertheless, kulokekahilide-2 and its derivatives are considered good anticancer candidates for testing in vivo efficacies in future human clinical trials. In the long run, further studies on specific mechanism of action, exploration of new derivatives as well as structural modifications would be of high significance for the development of kulokekahilide-2 as therapeutic agent for cancer treatment.

Kahalalide F is an intensively studied cyclic peptide that was first isolated from *Elysia rufescens* in 1993 [115]. Owing to its remarkable anticancer potential, several phase II clinical trials for kahalalide F are ongoing whilst a few had been completed [116]. As an analog of kahalalide F, kahalalide R_1_ (**71**) is also a cyclic depsipeptide, which was reported to be isolated from a species of sea slug *E. grandifolia* in 2006. Both kahalalide R_1_ and kahalalide F were tested for their cytotoxicities against MCF7 cells, and the results showed that they had comparable IC_50_ values of 0.14 and 0.22 µM, respectively. Furthermore, kahalalide R_1_ also exhibited anti-proliferative activity in mouse lymphoma L1578Y cell line with an IC_50_ value of 4.26 nM. The anti-proliferative potency of kahalalide R_1_ against cancer cells was almost the same potent as kahalalide F [117]. Nevertheless, further in vivo and preclinical studies on kahalalide R_1_ are urgently needed for a thorough validation of its anticancer potential.

### 2.4. Anticancer Cyclopeptides Derived from Marine Algae

In the plant kingdom, marine algae are the most ancient members responsible for maintaining the stability of the marine ecosystem though they are merely simple organisms possessing chlorophyll. Many bioactive compounds have also been derived from marine algae, therefore these microorganisms have attracted a lot of attention in the recent years for the exploration of algal products [118]. The structures of renowned bioactive algal compounds are provided in Figure 4 and the activities of these compounds are listed in Table 1.

Galaxamide (**72**) was a representative algae-derived cyclic pentapeptide discovered in marine algae *Galaxaura filamentosa*, which showed moderate inhibitory activities against GRC-1 (human renal cell carcinoma, IC_50_ = 7.18 μM) and HepG2 (IC_50_ = 7.81 μM) cell lines [119]. In the study of Lunagariya et al. [120], they investigated the mechanism of action of galaxamide on MCF7 cells in detail. Their results demonstrated that galaxamide induced apoptosis via the oxidative stress-mediated signaling pathway, by which the mitochondrial membrane potential was disrupted in response to the overwhelmed production of reactive oxygen species (ROS). Further, cell cycle arrest at G1 phase was also observed after the treatment with galaxamide due to ROS production, which contributed much to the apoptosis of MCF7 cells. Several synthetic analogs of galaxamide were found to be more potent than the natural scaffold, thus this type of compounds is worthy for further studies as novel candidates for the treatment of breast cancer.

**Table 1 ijms-22-03973-t001:** Marine-derived anticancer cyclopeptides.

Name	Biological Source	Anticancer Activity	Reference
Anticancer Cyclopeptides Derived from Sponge			
Azumamide A (**1**)	*Mycale izuensis*	HDAC inhibitory activity against K562 cells (IC_50_ = 0.045 μM); Cytostatic effects on WiDr (IC_50_ = 5.8 μM) and K562 (IC_50_ = 4.5 μM) cells	[35,36,37,38,39]
Azumamide E (**2**)	*Mycale izuensis*	HDAC inhibitory activity against K562 cells (IC_50_ = 0.045 μM)	[35,36,37,38,39]
Microsclerodermin A (**3**)	*Microscleroderma herdmani*	Induction of apoptosis in AsPC-1 (IC_50_ = 2.3 µM), BxPC-3 (IC_50_ = 0.8 µM) and PANC-1 (IC_50_ = 4.0 µM) cells.	[40,41,42]
Leucamide A (**4**)	*Leucetta microraphis*	Inhibitory activities to HM02 (GI_50_ = 8.5 µM), HepG2 (GI_50_ = 9.7 µM) and Huh7 (GI_50_ = 8.3 µM) cells.	[43,44]
Carteritin A (**5**)	Sponge, *Stylissa carteri*	Cytotoxicity against HCT116 (IC_50_ = 1.3 µM), RAW264 (IC_50_ = 1.5 µM) and HeLa (IC_50_ = 0.7 µM) cells	[45]
Stylissatin B (**6**)	*Stylissa massa*	Inhibitory effects on HCT116, HepG2, BGC823, NCI-H1650, A2780 and MCF7 cells. (IC_50_ = 2.3 to 10.6 µM)	[46]
Cupolamide A (**7**)	*Theonella cupola*	Cytotoxicity against P388 (IC_50_ = 7.5 µM) cell.	[47]
Geodiamolide H (**8**)	*Geodia corticostylifera*	Anti-proliferative activitiy against T47D (EC_50_ = 38.36 nM) and MCF7 (EC_50_ = 89.96 nM) cells	[48,49,50]
Theopapuamide (**9**)	*Theonella swinhoei*	Cytotoxicity against CEM-TART (EC_50_ = 0.5 µM) and HCT116 (EC_50_ = 0.9 µM) cells.	[51]
Homophymines A-E (**10**–**14**) & A1-E1 (**15**–**19**)	*Homophymia* sp.	Anti-proliferative against several cancer cell lines (IC_50_ = 2 to 100 nM), among which PC3 and OV3 are the most sensitive.	[52,53]
Pipestelide A (**20**)	*Pipestela candelabra*	Cytotoxicity against KB (IC_50_ = 0.1 µM) cell.	[54]
Neamphamides B-D (**21**–**23**)	*Neamphius huxleyi*	Cytotoxicity against A549, LNCaP and PC3 cells. (IC_50_ = 91 to 230 nM)	[55,56]
Callipeltins N (**24**) and O (**25**)	*Asteropus* sp.	Cytotoxicity against A2058, HT29, MCF7 and MRC-5 cells. (IC_50_ = 0.16 to 0.21 µM and 0.48 to 2.08 µM, respectively)	[57]
Pipecolidepsins A (**26**) and B (**27**)	*Homophymia lamellosa*	Cytotoxicity against A549 (GI_50_ = 0.6 and 0.04 µM, respectively), HT29 (GI_50_ = 1.12 and 0.01 µM, respectively) and MDA-MB-231 (GI_50_ = 0.7 and 0.02 µM, respectively) cells.	[58,59,60,61]
Halipeptin D (**28**)	*Leiosella* cf. *arenifibrosa*	Cytotoxicity against HCT116 cell line (IC_50_ = 7 nM) and BMS ODCP with an average IC_50_ value of 420 nM.	[62,63]
Reniochalistatin E (**29**)	*Reniochalina stalagmitis*	Cytotoxicity against RPMI-8226 (IC_50_ = 4.9 μM) and MGC-803 (IC_50_ = 9.7 μM) cells.	[64,65]
Callyaerin G (**30**)	*Callyspngia aerizusa*	Cytotoxicity against L5178Y (ED_50_ = 4.1 mM) and HeLa (ED_50_ = 41.8 mM) cells.	[66]
Callyaerins E (**31**) and H (**32**)	*Callyspngia aerizusa*	Cytotoxicity against L5178Y (ED_50_ = 0.39 and 0.48 μM, respectively).	[67]
Scleritodermin A (**33**)	*Scleritoderma nodosum*	Cytotoxicity against HCT116 (IC_50_ = 1.9 μM), HCT116/VM46 (IC_50_ = 5.6 μM), A2780 (IC_50_ = 0.94 μM) and SKBR3 (IC_50_ = 0.67 μM) cells.	[68,69,70]
Calyxamides A (**34**) and B (**35**)	*Discodermia calyx*	Cytotoxicity against P388 (IC_50_ = 3.9 and 0.9 μM, respectively) cell.	[71]
Keramamide E (**36**)	*Theonella* sp.	Cytotoxicity against L1210 (IC_50_ = 1.42 µM) and KB (IC_50_ = 1.38 µM) cells.	[72]
Keramamides K (**37**) and L (**38**)	*Theonella* sp.	Cytotoxicity against L1210 (IC_50_ = 0.77 and 0.5 µM, respectively) and KB (IC_50_ = 0.45 and 0.97 µM, respectively) cells.	[73]
Keramamides M (**39**) and N (**40**)	*Theonella* sp.	Cytotoxicity against L1210 (IC_50_ = 2.02 and 2.33 µM, respectively) and KB (IC_50_ = 5.05 and 6.23 µM, respectively) cells.	[74,75]
Axinellins A (**41**) and B (**42**)	*Axinella carteri*	Cytotoxicity against NSLC-N6 (IC_50_ = 16.7 and 29.8 µM, respectively) cell.	[76,77]
Stylopeptide 2 (**43**)	*Stylotella* sp.	Inhibition of 23% BT-549 cell growth and 44% Hs578T cell growth at one dose (10^−5^ M).	[78]
Stylissamide X (**44**)	*Stylissa* sp.	Anti-migration effects on HeLa cell.	[79,80]
Aciculitins A-C (**45**–**47**)	*Aciculites orientalis*	Cytotoxicity against HCT116 (IC_50_ = 0.37 µM) cell.	[81]
Nazumazoles A-C (**48**–**50**)	*Theonella swinhoei*	Cytotoxicity against P388 (IC_50_ = 0.83 µM) cell.	[82]
Theonellamide G (**51**)	*Theonella swinhoei*	Cytotoxicity against HCT116 (IC_50_ = 6.0 µM) cell.	[83]
Koshikamide B (**52**)	*Theonella* sp.	Cytotoxicity against HCT116 (IC_50_ = 3.62 µM) and P388 (IC_50_ = 0.22 µM) cells.	[84]
Anticancer Cyclopeptides Derived from Ascidians/Tunicates			
Mollamide (**53**)	Ascidian, *Didemnum molle*	Cytotoxicity against P388 (IC_50_ = 1.24 µM), A549 (IC_50_ = 3.1 µM), HT29 (IC_50_ = 3.1 µM) and CV1 (IC_50_ = 3.1 µM) cells.	[87,88]
Mollamide B (**54**)	Tunicate, *Didemnum molle*	Growth inhibition of H460, MCF7 and SF-268 cells.	[89]
Trunkamide A (**55**)	Ascidian, *Lissoclinum* sp.	Cytotoxicity against A549, P388, HT29 and MEL28 cells. (IC_50_ = 0.6 to 1.19 µM)	[90,91]
Prepatellamide A (**56**)	Ascidian, *Lissoclinum patella*	Cytotoxicity against P388 cells. (IC_50_~6.57 µM)	[92]
Vitilevuamide (**57**)	Ascidian, *Didemnum cuculiferum*	Cytotoxicity against HCT116 (IC_50_ = 6.24 nM), A549 (IC_50_ = 0.12 µM), SKMEL-5 (IC_50_ = 0.31 µM) and A498 (IC_50_ = 3.12 µM) cells.	[93,94]
Tamandarin A (**58**)	Ascidian, *Didemnidae* sp.	Cytotoxicity against BX-PC3 (IC_50_ = 1.69 nM), DU145 (IC_50_ = 1.29 nM), UM-SCC-10B (IC_50_ = 0.94 nM) cells.	[95,96,97]
Comoramides A-B (**59**–**60**) & Mayotamides A-B (**61**–**62**)	Ascidian, *Didemnum molle*	Cytotoxicity against A549, HT29 and MEL28 cells. (IC_50_ = 7.22 to 14.97 µM)	[98]
Patellin 6 (**63**)	Ascidian, *Lissoclinum patella*	Cytotoxicity against P388, A549, HT29 and CV1 cells. (IC_50_~2.08 µM)	[99]
Cycloforskamide (**64**)	Sea slug, *Pleurobranchus forskalii*	Cytotoxicity against P388 cell (IC_50_ = 5.8 µM).	[100]
Anticancer Cyclopeptides Derived from Mollusks			
Keenamide A (**65**)	Mollusk, *Pleurobranchus forskali*	Anti-proliferative activity against A549, P388, MEL20 and HT29 cells. (IC_50_ = 4.03 to 8.05 µM)	[103]
Aurilide (**66**)	Sea hare, *Dolabella auricularia*	Cytotoxicity against HeLa S_3_ (IC_50_ = 0.013 µM) cell. Inhibitory activity against ovarian, renal and prostate cancer cells in NCI 60 cell lines.	[104,105]
Dolastatin 16 (**67**)	Sea hare, *Dolabella auricularia*	Cytotoxicity against H460 (GI_50_ = 1.09 nM), KM20L2 (GI_50_ = 1.37 nM), SF-295 (GI_50_ = 5.92 nM) and SKMEL-5 (GI_50_ = 3.75 nM) cells. Anti-proliferative effects against 5 human leukemia cell lines	[106]
Doliculide (**68**)	Sea hare, *Dolabella auricularia*	Cytotoxicity against HeLa S_3_ (IC_50_ = 1.62 nM) cell.	[107,108,109,110]
Kulokekahilide-1 (**69**)	Mollusk, *Philinopsis specio**sa*	Cytotoxicity against P388 (IC_50_ = 2.2 µM) cell.	[111]
Kulokekahilide-2 (**70**)	Mollusk, *Philinopsis speciosa*	Cytotoxicity against P388 (IC_50_ = 4.2 nM), SK-OV-3 (IC_50_ = 7.5 nM), MDA-MB-435 (IC_50_ = 14.6 nM) and A-10 (IC_50_ = 59.1 nM) cells.	[112,113,114]
Kahalalide R_1_ (**71**)	Sea slug, *Elysia grandifolia*	Cytotoxicity against MCF7 (IC_50_ = 0.14 µM), L1578Y (IC_50_ = 4.26 nM) cells.	[117]
Anticancer Cyclopeptides Derived from Marine Algae			
Galaxamide (**72**)	Algae, *Galaxaura filamentosa*	Inhibitory activity against GRC-1 (IC_50_ = 7.18 μM) and HepG2 (IC_50_ = 7.81 μM) cells.	[119,120]

## 3. Terrestrial Plant-Derived Anticancer Cyclopeptides 

Plants are undeniably an important source for natural products. Nowadays, traditional plant-based medicines are still prevalently used for healthcare and medical treatment around the world. Over 28,000 species of plants have been recorded with differenct therapeutic purposes [121]. Therefore, plants constituents have long been regarded as the mainstream materials for drug discovery. In particular, over one third of current anticancer drugs are derived from plant natural products [122]. In this section, the recently isolated plant-derived cyclopeptides and their anticancer properties are discussed. The structures of these compounds are shown in Figure 5 and their activities are listed in Table 2.

Cherimolacyclopeptides C-F (**73**–**76**) were cyclic heptapeptides obtained from the seeds of *Annona cherimola* in 2004 and 2005. All four compounds exhibited significant in vitro cytotoxicity against KB cells with IC_50_ values ranging from 0.017 to 0.97 μM [123,124,125]. Although these compounds showed anticancer potential, no further screening on other cancer cell lines or pharmacological investigations of these congeners have been reported thereafter.

Integerrimide C (**77**), a novel cycloheptapeptide, has been recently isolated from the latex of *Jatropha integerrima*. The cytotoxicity bioassay indicated its inhibitory effect against KB cells was associated with an IC_50_ value of 1.7 μM [126]. The discovery of integerrimide C was actually the continuation of studies on the latex of *J. integerrima*. Previously reported cyclopeptides integerrimides A and B were also the isolates of this particular species and exhibited moderate inhibitory activities against human melanoma (IPC-298) cell proliferation (up to 40% at 50 μM) and human pancreatic carcinoma (Capan II) cell migration (30% and 20% at 50 μM, respectively) [127].

Linoorbitides (LOBs) are a group of cyclopeptides found in flaxseed oil. Actually, flaxseed and its oil have been considered and utilized as anticancer products for many years [128]. In order to investigate whether flaxseed-derived LOBs are cytotoxic to cancer cells, Denis et al. [129]. tested the cytotoxicity of four cyclopeptides derived from flaxseed against A375 (melanoma), SKBR3 and MCF7 cells. Their study showed that LOB3 (**78**) possessed the highest in vitro potency among the test compounds, yet the minimum concentration of LOBs in serum had to be reached about 400–500 μg/mL for the delivery of its effectiveness. Such concentration is unlikely to be used as a drug through oral administration in humans. Nevertheless, as topical medication, LOB3 could be a potential agent for the treatment of melanoma and other skin cancers. Although flaxseed is well known for its health benefits against cancer, searching for specific compounds derived from flaxseed with high potency is still under way.

Cambodine A (**79**) was a novel 14-membered ring cyclopeptide extracted from the root barks of *Ziziphus cambodiana*. The in vitro bioassay showed that it was moderately cytotoxic against the BC-1 (lymphoma) cells with an IC_50_ value of 11.1 µM, whilst no toxicity was shown against the non-cancerous Vero cells [130].

The species *Dianthus superbus* is an important anti-inflammatory and diuretic herb in traditional Chinese medicine. Longicalycinin A (**80**), a cytotoxic cyclopeptide, was first isolated from the extract of *D. superbus* several years ago. The natural compound showed a growth inhibitory activity against HepG2 cancer cells with an IC_50_ value of 22.13 µM [131]. This compound had been successfully synthesized by the solid-phase methodology. When acting against in vitro Dalton’s lymphoma ascites (DLA) and Enrlich’s ascites carcinoma (EAC) cells, CTC_50_ values were determined to be 2.62 and 6.17 µM, respectively [132]. In the study of Tehrani et al., they demonstrated that the synthesized linear and cyclic disulfide heptapeptides of longicalycinin A showed similar inhibitory effect against HepG2 (IC_50_ = 16.97 and 16.91 µM, respectively) and HT29 (IC_50_ = 20.38 and 27.68 µM, respectively) cells. However, considering their vulnerability and proapoptotic actions against normal cells (skin fibroblast cells), the linear disulfide heptapeptides were more tempting for promotion as novel anticancer agents [133]. Although longicalycinin A and its analogs can be totally synthesized, the mechanism of actions as well as the in vivo actions of these compounds are yet to be thoroughly studied.

Similar to longicalycinin A, dianthin E (**81**) is another cyclic peptide isolated from *Dianthus superbus*. With the IC_50_ values of >33.3 µM against Hep3B (human hepatic carcinoma), MCF7, A549 and MDA-MB-231 cells, this compound exhibited a selective in vitro cytotoxic activity against HepG2 cells (IC_50_ = 3.51 µM) [134].

Orbitides are a group of plant cyclic peptides possessing a signature short chain of 5 to 11 residues, they are biosynthesized by ribosomes and characterized by their N-to-C amide bonds, but not disulfide bonds [135]. [1-8-NαC]-Zanriorb A1 (**82**) was a novel orbitide isolated from the leaves of *Zanthoxylum riedelianum* with proapoptotic effects. This compound was found to be cytotoxic against Jurkat leukemia T cells with an IC_50_ value of 218 nM. Regarding its molecular mechanism, it induced remarkable cell death, which would be partially inhibited by the caspase inhibitor Z-VAD-FMK. Moreover, it also induced apoptosis via reducing the levels of mitochondrial membrane potential (Ψmit) and deactivating caspase-3 [136].

Three novel cyclopeptide alkaloids, justicianenes B-D (**83**–**85**) were isolated from the *Justicia procumbens* L. These three compounds were evaluated their cytotoxicity against human breast cancer MCF-7, cervix carcinoma HeLa, and lung cancer A549 and H460. However, only justicianene D displayed weak cyctotoxicity against MCF-7 cells with IC_50_ of 90 μM [137].

Seven novel ginseng cyclopeptides (GCPs) were isolated and showed anti-proliferative activities in gastric cancer SGC-7901 cells, in which GCP-1 (**86**) showed the most potency with IC_50_ value of 37.8 μM. Regarding its molecular mechanism, it induces apoptosis by activating the caspases and regulating thioredoxin (Trx)-dependent pathways, including signal-regulating kinase 1 (ASK1), mitogen-activated protein kinases (MAPKs)-p38 and JNK pathways [138].

Cyclotides are the macrocyclic cysteine-rich peptides derived from plants, and featured with three disulfide bonds, 28 to 37 amino acids and a head-to-tail cyclized backbone in a knotted arrangement [139]. The typical structure of cyclotides makes them broadly bioactive and remarkably stable against enzymatic and thermal degradation [140].

In traditional Chinese medicine, the roots and rhizomes of *Rubia* plants have been used for thousands of years to treat menoxenia, contusion, rheumatism and tuberculosis [141]. Rubiaceae-type cyclopeptides (RAs) are referred to as natural cyclopeptides derived from *Rubia*. Up to now, 56 RAs have been isolated from *Rubia* plants. Due to their characteristic bicyclic structures and significant anticancer activities, RAs have gained much attention in recent years [142]. Rubipodanin A (**87**), a novel cyclic hexapeptide obtained from the roots and rhizomes of *R. podantha*, is the first naturally identified *N*-desmonomethyl RA. This compound was tested to be cytotoxic against three cancer cell lines including HeLa, A549 and SGC-7901 (human gastric cancer cell), with IC_50_ values ranging from 3.80 to 7.22 μM. However, when compared to the previously known cyclopeptide RA-V, in which two *N*-methyl groups are present in its backbone skeleton, the cytotoxicity of rubipodanin A was considered much weaker. In regards to its mechanisms, rubipodanin A exhibited a down-regulating effect on the nuclear factor kappa-light-chain-enhancer of activated B cells (NF-κB) pathway, which largely contributed to its anticancer activity [143]. Derived from the same plant species, another new RA with potent cytotoxicity was designated rubipodanin B (**88**). This compound showed IC_50_ values of 1.47 μM, 0.69 μM and 3.37 μM in MDA-MB-231, SW620 (human colorectal adenocarcinoma) and HepG2 cells respectively. Nevertheless, the anticancer activity of rubipodanin B was not associated with any down-regulation of the NF-κB signaling pathway [144]. Another cyclopeptide compound from RA family is RA-XII (**89**), which derived from *R. yunnanensis*, showed inhibitory effects on tumor growth and metastasis IC_50_ value of 606 nM and 96 nM on human breast cancer 4T1 cells for 24 and 48 h, respectively [145,146,147]. RA-XII also exerts antitumor activity by supressing autophagy via activating Akt-mTOR and NF-κB pathways in colocrectal cancer SW620 and HT29 cells [148]. RA-XXV (**90**) and RA-XXVI (**91**) are two new bicyclic hexapeptides, which isolated from the roots of R. cordifolia L. RA-XXV showed cytotoxic activity against human promyelocytic leukemia HL-60 and human colorectal carcinoma HCT 116 cells with IC_50_ of 62 nM and 28 nM, respectively. RA-XXVI showed cytotoxic activity agains HL-60 and HCT 116 cells with with IC_50_ of 66 nM and 51 nM, respectively [149]. To further determine RA compounds underlying mechanism, a wide screening of its molecular targets is desperately needed.

The intensive studies of famous traditional Chinese herb *Hedyotis biflora* had led to the isolation of HB7 (**92**), which is considered a potential anticancer cyclotide. By means of MTT assay, HB7 showed in vitro cytotoxicity against four pancreatic cancer cell lines BxPC3, Capan2, MOH-1 and PANC1 with IC_50_ values of 0.68, 0.45, 0.33 and 0.36 µM, respectively. At a non-toxic concentration of 0.05 µM, HB7 even inhibited cellular migration and invasion of Capan2 cells. When extrapolated to the in vivo xenograft mouse models, this cyclotide was observed to significantly suppress tumor growth without exhibiting any obvious organ injury or toxicity. Based on the results above, the cytotoxicity of HB7 was plausibly related to the different net charges of the cyclotide [150]. Because of its decent anticancer potential, the specific underlying mechanism of action of HB7 deserves further investigation.

Cliotides T1-T4 (**93**–**96**) were four novel cyclotides isolated from the tropical plant *Clitoria ternatea*, belonging to the Fabaceae family. With the prevalent distribution in almost every tissue of *C. ternate*, the extraction of these four cyclotides is rather efficient. These cyclotides have been confirmed to be heat-stable cysteine-rich peptides. When incubated with HeLa cells, all these four compounds yielded significant cytotoxicity with IC_50_ values ranging from 0.6 to 8.0 µM. Among the four, T1 and T4 were the most potent ones (IC_50_ = 0.6 µM) [151]. Cliotides T2 (**94**), T4 (**96**), T7 (**97**), T10 (**98**) and T12 (**99**) also exhibited significant cytotoxicity towards A549 cells (IC_50_ values ranging from 0.21 to 7.59 µM) as well as paclitaxel-resistant A549 cells (IC_50_ values ranging from 0.45 to 7.92 µM). Intriguingly, the IC_50_ values of these five test compounds were decreased by two to four folds when incubated with paclitaxel [152]. Nevertheless, the chemosensitizing ability of the cliotides became evidenced in drug-resistant cell lines, which deserves further studies on the aspects of compound charge status and in vivo efficacy. Apart from anticancer potential, cliotides, T1 to T4 in particular, also exhibited notable antimicrobial properties, particularly *E. coli* [152].

Among the cyclotides isolated from *Psychotria leptothyrsa* var. *longicarpa*, psyle E (**100**) was the most cytotoxic one. The IC_50_ value of psyle E against human lymphoma cell line U937-GTB was determined to be 0.76 µM. According to the reported SAR assessment, the linear structure of psyle E was required to maintain its potent cytotoxicity [153].

Vigno 5 (**101**) was a cyclotide discovered in *Viola ignobilis* [154]. This compound showed its in vitro cytotoxicity against HeLa cells with an IC_50_ value around 2.5 µM. Due to the complexity of its chemical structure and technical limitation, only the primary structure of vigno 5 had been elucidated so far. In the study of Esmaeili el al., vigno 5 induced apoptosis in a dose-dependent manner accompanied by nuclear shrinkage, DNA fragmentation, caspase activation and the cleavage of PARP. The vigno 5 induced-apoptosis was found to be caspase-dependent, explicitly caspase-3. To the induction of apoptotic events and mitochondrial dysfunction, a decreased level of anti-apoptotic Bcl-2 and an increased level of pro-apoptotic Bax were observed in the vigno 5-treated HeLa cells [155].

Cycloviolacin O2 (CyO2, **102**), a potent cytotoxic cyclotide, was first isolated from the plant *Viola odorata* L. (Violaceae). The presence of glutamic acid in cycloviolacin O2 contributed much to its cytotoxic activity against U937 GTB (human lymphoma) cells with an IC_50_ value of 0.75 μM [156]. Besides, CyO2 also exhibited concentration-dependent cytotoxicities towards CCRF-CEM (leukemia cell line), NCI-H69 (small cell lung cancer) and HT29 cells with IC_50_ values of 0.3, 1.2 and 5.3 μM, respectively [157]. Disruption of the U937 cell membranes by CyO2 further indicated its membrane-disrupting activity while exerting cytotoxicity [158]. Gerlach et al. even demonstrated that CyO2 induced pore formation specifically in highly proliferating tumor cells [159]. In addition, CyO2 was also reported to exhibit bioactivities against gram-negative bacteria and HIV-1 virus via multipleregulatory mechanisms [153,160]. However, another study on the evaluation of toxicity and antitumor activity of CyO2 in mice showed that its antitumor effects were little or even absent at sublethal doses. It is worth noting that quick lethality of CyO2 was observed at 2 mg/kg whilst no abnormal signs were seen at the dose of 1.5 mg/kg in the experimental mice [157]. Thus, there is a great disparity between the results of the in vitro and in vivo experiments. For further investigation of this cyclotide, problems such as low in vivo efficacy need to be addressed.

Characterized by the macrocyclic backbones, cyclotides viphis A-G (**103**–**109**) were isolated from *Viola philippica*. These eight compounds were cytotoxic to human melanoma (MM96L), HeLa and human gastric adenocarcinoma (BGC-823) cells and human normal fibroblast cells (HFF-1). When applied at 3.24 and 3.17 µM, viphis D-E showed no activity against BGC-823 cell; however all these eight cyclotides displayed cytotoxicities against many cancer cell lines with IC_50_ values ranging from 1.03 to 15.5 µM. From the results of SAR assessments, hydrophobicity and glutamic acid residue of loop 1 are suggested of great importance for their cytotoxic activities. Further biochemical assays revealed that these viphis could be influenced by minimal sequential changes for alterations of their bioactivities [161].

Vabys A (**110**) and D (**111**) were two cyclotides derived from *Viola abyssinica*, a plant growing at an altitude over 3400 m. Both compounds contain charged residues; however different net charges lead to different bioactivities. Vabys A and D exhibited cytotoxicities against U-937 lymphoma cells in a dose-dependent manner, with IC_50_ values of 2.6 and 7.6 µM, respectively [162].

Isolated from the alpine violet *Viola biflora*, vibis G (**112**) and H (**113**) were two bracelet cyclotides. These two compounds showed similar cytotoxic potency (IC_50_ = 0.96 and 1.6 µM, respectively) against lymphoma U-937 GTB cells [163].

A bioactivity-guided fractionation of *Viola tricolor* led to the isolation of a cluster of highly similar cyclotides, among which vitri A (**114**), varv A (**115**) and varv E (**116**) were found to be cytotoxic agents. Two human cancer cell lines, U-937 GTB (IC_50_ = 0.6, 6 and 4 µM, respectively) and myeloma RPMI-8226/s (IC_50_ = 1, 3 and 4 µM, respectively) were used in the cytotoxicity assays. Sequence determination and SAR analysis demonstrated that differences in the net charges and cationic amino acid residues in these cyclotides are extremely crucial for their cytotoxicities [164]. Assessment of the cytotoxic activities against human malignant glioblastoma (U251), MDA-MB-231, A549, DU145 and human hepatoma (BEL7402) cell lines showed the inhibitory activities of vitri A (**114**) with IC_50_ values ranging from 0.97 to 1.91 µM and vitri F (**117**) with IC_50_ values ranging from 0.85 to 1.97 µM. Importantly, the distribution of highly hydrophobic residues on the surface was highlighted for the cytotoxic activities of these cyclotides [165]. Undeniably, plant-derived cyclotides are the intriguing candidates for drug development as anticancer therapeutics. Owing to the abundant source of cyclotides, *V. tricolor* warrants further studies for the identification of new bioactive compounds and their anticancer mechanism of actions.

**Table 2 ijms-22-03973-t002:** Terrestrial plant-derived anticancer cyclopeptides.

Name	Biological Source	Anticancer Activity	Reference
Cherimolacyclopeptides C-F (**73**–**76**)	Seeds of *Annona cherimola*	Cytotoxicity against KB cell. (IC_50_ = 0.017 to 0.97 µM)	[123,124,125]
Integerrimide C (**77**)	Latex of *Jatropha integerrima*	Cytotoxicity against KB (IC_50_ = 1.7 µM) cell.	[126]
LOB3 (**78**)	Flaxseed oil	Cytotoxicty against A375, SKBR3 and MCF7 cells.	[128,129]
Cambodine A (**79**)	Root bark of *Ziziphus cambodiana*	Cytotoxicity against BC-1 (IC_50_ = 11.1 µM) cell.	[130]
Longicalycinin A (**80**)	*Dianthus superbus*	Growth inhibitory activity against HepG2 (IC_50_ = 22.13 μM) cell.	[131,132,133]
Dianthin E (**81**)	*Dianthus superbus*	Cytotoxicity against HepG2 (IC_50_ = 3.51 μM) cell.	[134]
[1-8-NαC]-Zanriorb A1 (**82**)	Leaves of *Zanthoxylum riedelianum*	Cytotoxicity against Jurkat leukemia T cell (IC_50_ = 218 nM).	[136]
Justicianenes B-D (**83**–**85**)	*Justicia procumbens* L.	Justicianene D displayed cytotoxicity against MCF-7 cells (IC_50_ = 90 μM).	[137]
GCP-1 (**86**)	Ginseng	Cytotoxicity against SGC-7901 cells (IC_50_ = 37.8 μM).	[138]
Rubipodanin A (**87**)	Roots and rhizomes of *Rubia podantha*	Cytotoxicity against HeLa, A549 and SGC-7901 cells. (IC_50_ = 3.80 to 7.22 μM)	[143]
Rubipodanin B (**88**)	*Rubia podantha*	Cytotoxicity against MDA-MB-231 (IC_50_ = 1.47 μM), SW620 (IC_50_ = 0.69 μM) and HepG2 (IC_50_ = 3.37 μM) cells.	[144]
RA-XII (**89**)	*Rubia yunnanensis*	Inhibitory effects on 4T1 ((IC_50_ = 606 nM and 96 nM for 24 and 48 h, respectively), SW260 and HT29 cells.	[145,146,147]
RA-XXV (**90**)	Roots of *Rubia cordifolia* L.	Cytotoxicity against HL-60 ((IC_50_ = 62 nM) and HCT116 cells ((IC_50_ = 28 nM).	[149]
RA-XXVI (**91**)	Roots of *Rubia cordifolia* L.	Cytotoxicity against HL-60 ((IC_50_ = 66 nM) and HCT116 cells ((IC_50_ = 51 nM).	[149]
HB7 (**92**)	*Hedyotis biflora*	Cytotoxicity against BxPC3 (IC_50_ = 0.68 μM), Capan2 (IC_50_ = 0.45 μM), MOH-1 (IC_50_ = 0.33 μM) and PANC1 (IC_50_ = 0.36 μM) cells.	[150]
Cliotides T1-T4 (**93**–**96**), cliotide T7 (**97**), cliotide T10 (**98**), cliotide T12 (**99**)	*Clitoria ternatea*	Cliotides T1-T4: cytotoxicity against HeLa (IC_50_ = 0.6 to 8.0 μM) cell. Cliotides T2, T4, T7, T10 and T12: cytotoxicity against A549 (IC_50_ = 0.21 to 7.59 μM) and A549/paclitaxel (IC_50_ = 0.45 to 7.92 μM) cells.	[151,152]
psyle E (**100**)	*Psychotria leptothyrsa* var. *longicarpa*	Cytotoxicity against U937-GTB IC_50_ = 0.76 μM) cell.	[153]
Vigno 5 (**101**)	*Viola ignobilis*	Pro-apoptotic activity on HeLa cell.	[154,155]
Cycloviolacin O2 (CyO2) **(102**)	*Viola odorata* L.	Cytotoxicity against U937 GTB (IC_50_ = 0.75 μM), CCRF-CEM, NCI-H69 and HT29 cells.	[156,157,158,159,160]
Viphi A-G (**103**–**109**)	*Viola philippica*	Cytotoxicity against MM96L, HeLa, BGC-823, HFF-1 cells, but viphi D and E showed no activity against BGC-823 cell. (IC_50_ = 1.03 to 7.92 μM)	[161]
Vaby A (**110**) and D (**111**)	*Viola abyssinica*	Cytotoxicity against U-937 (IC_50_ = 2.6 and 7.6 μM, respectively) cell.	[162]
Vibi G (**112**) and H (**113**)	*Viola biflora*	Cytotoxicity against U-937 GTB (IC_50_ = 0.96 and 1.6 μM, respectively) cell.	[163]
Vitri A (**114**)	*Viola tricolor*	Cytotoxicity against U-937 GTB (IC_50_ = 0.6 µM) and RPMI-8226/s (IC_50_ = 1 µM) cells. Cytotoxicity against U251, MDA-MB-231, A549, DU145 and BEL7402 cells (IC_50_ = 3.07 to 6.03 μM).	[164,165]
Varv A (**115**) and E (**116**)	*Viola tricolor*	Cytotoxicity against U-937 GTB (IC_50_ = 6 and 4 µM, respectively) and RPMI-8226/s (IC_50_ = 3 and 4 µM, respectively) cells.	[164]
Vitri F (**117**)	*Viola tricolor*	Cytotoxicity against U251, MDA-MB-231, A549, DU145 and BEL7402 cel ls (IC_50_ = 2.74 to 6.31 μM).	[165]

## 4. Bacteria-Derived Anticancer Cyclopeptides

Bacteria are single celled micro-organisms that thrive in diverse environments. In fact, a significant number of bacteria are from oceans. It is hard to imagine that there are actually 40 million bacterial cells living in 1 g of soil and 1 million bacterial cells in 1 mL of freshwater [166]. Diverse habitats and complex relationships with other organisms, such as antagonistic actions towards some worms and molecules, cause bacteria to produce many different types of metabolites, and some of them are highly bioactive for pharmaceutical uses in humans [167]. In recent years, numerous bioactive cyclopeptides have been isolated from bacteria possessing anticancer properties. In this section, these cyclopeptides and their cytotoxicities against cancer cells are discussed. The chemical structures of these cyclopeptides are provided in Figure 6 and their activities were listed in Table 3.

### 4.1. Anticancer Cyclopeptides Derived from Non-Marine Bacteria

YM753 (spiruchostatin A/OBP-801) (**118**), a disulfide bond-containing bicyclic depsipeptide, was isolated from *Pseudomonas* sp. and identified as a gene-enhancing substance at first. Subsequent studies also demonstrated it also exerted TGF-β-like activity and HDAC inhibitory properties [168,169]. As a HDAC inhibitor, YM753 induced histone acetylation through cell cycle arrest at the G1 and G2/M phases and enhancing p21^WAF1/Cip1^ expression and apoptosis in WiDr cells [169]. When applied with celecoxib as a combo treatment, a synergetic anti-proliferative effect was observed in the high-grade bladder cancer cells. Such anti-proliferation of cancer cells was found to be caspase dependent in both in vitro and in vivo experiments [170]. The combinatory treatment with OBP-801 and fibroblast growth factor receptor (FGFR) inhibitor BGJ298, a significant attenuation on cell growth was obtained in the muscle-invasive bladder cancer cells, and a regulation of the caspase and Bim pathways was demonstrated [171]. Accordingly, these evidences have demonstrated the great anticancer potential of combo remedies YM753/OBP-801/spiruchostatin A.

Telomestatin (**119**) was a macrocyclic peptide first isolated from *Streptomyces anulatus* 3533-SV4 and had been shown to act as a telomerase inhibitor in cancer cells [172]. Telomerase is indeed a significant enzyme responsible for cellular senescence and tumorigenic conversion. Telomestatin can specifically bind to the telomeric G-quadruplex structure at the 3′ telomere end and inhibit the activity of telomerase [173]. When acting against MiaPaCa (human pancreatic carcinoma) cells, the IC_50_ value of telomestatin was determined to be 0.5 μM [174]. This compound also exhibited significant cytotoxicity against four neuroblastoma cell lines with IC_50_ values ranging from 0.8 to 4.0 μM. Telomere shortening, cell cycle arrest and apoptosis were observed after long-term incubation with telomestain at non-cytotoxic concentrations [175]. Due to its potent cytotoxicity against cancer cells, telomestatin is considered as a potential therapeutic agent in treating cancers [176]. The pharmacological characteristics of telomestatin promotes it to be served as a reference standard substance in the in vitro telomere shortening activity assays [177]. Although a great number of telomerase inhibitors have been isolated from the nature, telomestatin remains the strongest cytotoxic agent of its kind. However, the limited production has impeded the wide uses of telomestatin. Fortunately, the total synthesis of telomestatin was achieved by the coupling reactions of cysteine-containing trisoxazole amine and serine-containing trisoxazole carboxylic acid in 2006 [178]. In a research reported in 2017, the biosynthesis of telometstatin was further proposed and carried out using a cluster of genes by assembling the amino acids cysteine, serine and threonine, which may be provide a promising approach for mass production of this compound for future preclinical and clinical trials [176].

### 4.2. Anticancer Cyclopeptides Derived from Marine Bacteria

As the smallest autonomous organisms in the ocean, marine bacteria can exist independently or symbiotically with other marine microorganisms [177,179]. Because marine bacteria are always the excellent origins of natural therapeutic agents, they are described as the chemical gold [180]. Numerous compounds, including cyclopeptides, with anticancer activities have been extracted from marine bacteria. The chemical structures of these cyclopeptides are provided in Figure 7 and their activities were listed in Table 3.

Thiocoraline (**120**) was a potential antitumor cyclic depsipeptide produced by the the Gram-positive bacteria *Micromonospora* sp. ACM2-092 and *Micromonospora* sp. ML1 [181]. This compound exerted cytotoxicities against P388, A549, MEL28 and HT29 cells with IC_50_ values of 1.72, 1.72, 1.72 and 8.64 nM, respectively [182]. By using the human colorectal adenocarcinoma LoVo and SW620 cells as the cellular models, thiocoraline was found to induce cell cycle arrest at the G1 phase. Its anticancer effects may also be related to its inhibition of DNA polymerase α activities [183]. To the human pancreatic carcinoid BON and pulmonary carcinoid H727 cells, the thiocoraline treatment led to increased expression levels of poly ADP ribose polymerase (PARP) and X-linked inhibitor of apoptosis protein (XIAP), indicating that the anti-proliferative activity of this compound was highly associated with the induction of apoptosis [184]. Furthermore, in the aggressive neuroendocrine tumor medullary thyroid cancer MTC-TT cells, the IC_50_ value of the compound was determined to be 7.6 nM. The studies on its mechanism of action demonstrated that thiocoraline induced induce cell cycle arrest at the G1 phase and activated the NOTCH pathway [185]. Overall, the findings above illustrated that thilcoraline is worthy for further investigation to reveal its anticancer potential as a palliative therapeutic agent.

Ohmyungsamycins A (**121**) and B (**122**) were two new cyclodepsipeptides purified from the marine bacteria of *Streptomyces* sp. of a volcanic island area in Korea. Both cyclodepsipeptides showed selective cytotoxicity towards cancer cells over normal cells as no notable cytotoxicity (IC_50_ > 40 µM) against normal lung epithelial cells (MRC-5) was observed. When applied to the HCT116, A549, SNU-638 (human gastric adenocarcinoma), MDA-MB-231 and SK-HEP-1 (human liver adenocarcinoma) cells, ohmyungsamycin A (IC_50_ values ranging from 359 to 816 nM) exhibited inhibitory activities much higher than ohmyungmycin B (IC_50_ values ranging from 12.4 to 16.8 µM). The only structural difference between these two compounds is that the *N*-methylvaline moiety on the side chain of ohmyungmycin A is replaced by the *N*,*N*-dimethylvaline moiety in ohmyungmycin B. In the CRC HCT116 cells, ohmyungmycin A was found 78 times more potent than the positive reference drug etoposide. On the other hand, ohmyungmycin A also displayed much better antibacterial activities than ohmyungmycin B. Collectively, the *N*-methyl group at the terminus of ohmyungmycin B is a hinder to its bioactivities [186]. As total syntheses of ohmyungmycins A and B had been established, investigations of these two compounds could be comprehensively performed. A few reports even suggested that these two cyclopeptides deliver antituberculosis properties [187]. Therefore, studies on synthesizing their analogs with higher selectivity and bioactivities and exploring their mechanisms of actions are inspired.

Arenamides A (**123**) and B (**124**) were two novel cyclohexadepsipeptides isolated from the marine bacterial strain *Salinispora arenicola*. These two compounds merely exhibited moderate in vitro cytotoxicity against HCT116 cells with IC_50_ values of 19.7 and 29.8 µM, respectively. However, arenamides A and B were able to block tumor necrosis factor (TNF)-induced activities at rather low concentrations (IC_50_ = 5.51 and 2.64 µM, respectively). When the lipopolysaccharides (LPS)-challenged RAW 264.7 macrophages were treated with arenamides A or B, the generation of nitric oxide (NO) and prostaglandin E2 (PGE2), so as the major oxidative mediators of the NFκB pathway, were significantly inhibited whereas cell viability was not influenced. The above data suggested these two compounds carry chemopreventive potential [188]. The successful total synthesis of arenamide A has been reported [189]. which provided an alternative approach to obtain this compound as well as to synthesize its anologues.

Piperazimycins A-C (**125**–**127**) were three cytotoxic hexadepsipeptides isolated from the genus *Streptomyces* living in the marine sediment near the island of Guam. They showed in vitro cytotoxicities against HCT116 accompanied with an average GI_50_ value of 0.1 µM. Such low value unveiled the significant anticancer potency of this type of compounds. When screened against the NCI 60 cancer cell line panel, piperazimycin A exhibited a mean GI_50_ value of only 100 nM. Although piperazimycin A was shown with a selective growth inhibition against solid tumor cell lines over leukemia cell lines, the general mode of cytotoxicity was observed in the NCI cell line panel assay [190]. Further comprehensive studies on piperazimycin A can be proceeded as the total synthesis of piperazimycin A had been accomplished [191].

Obtained from a piece of driftwood collected at Kailua beach in Oahu, a Gram-negative bacterium BH-107 yielded four cyclic acyldepsipeptides kailuins A-D (**128**–**131**). These compounds represent a new type of lipopeptides with five amino acids and a β-acyloxy fatty acid moiety arranged in a 19-membered ring. All these four compounds displayed inhibitory activities against A549, MCF7 and HT29 cells with GI_50_ values ranging from 2.66–5.52 µM [192].

Mixirins A-C (**132**–**134**) were three cyclic acylpeptides obtained from marine bacteria of the *Bacillus* genus. The anti-proliferative activities of these three compounds were evaluated in HCT116 cells whereas their IC_50_ values were determined to be 0.65, 1.6 and 1.26 µM, respectively [193].

Four cyclopeptides were isolated from the ethyl acetate extract of *Bacillus velezensis* sp. RA5401. All four compounds showed anti-proliferative effects against human breast cancer MDA-MB-231 cells. Their molecular-docking studies showed that some G-protein coupled receptors (GPCRs) could be the potential binding targets. However, the IC_50_ of these four compounds and the mechanism of actions against cancer cells remain unknown and are worthy for further studies [194].

*Bacillus* sp. SY27F collected from the Indian Ocean hydrothermal vent yielded bacilohydrin A (**135**), a novel cyclic lipopeptide with significant anticancer potential. Among the six tested cancer cell lines, bacilohydrin A exhibited rather high cytotoxicity against the HepG2, MCF7 and DU145 cells, with IC_50_ values varying from 50.3 to 175.1 nM. In addition to its cytotoxic effect, cell shrinkage, condensation and DNA fragmentation were also observed as the result of apoptosis [195]. More studies are therefore needed to explore its detailed mechanism of action and therapeutic potential for cancer treatment.

Dolyemycins A (**136**) and B (**137**) are cyclopeptides isolated from *Streptomyces griseus* subsp, *griseus* HYS31 by bio-guided isolation. Both compounds showed anti-proliferative activities against human lung cancer A549 cells with IC_50_ values of 1.0 and 1.2 µM, respectively [196].

Sungsanpin (**138**) is a lasso peptide that contains a cyclic peptide and a tail. This compound was first isolated from the bacteria of *Streptomyces* sp. In the deep-sea sediment. In the cell invasion assay of A549 cells, the invading activity of the cells was significantly inhibited by sungsanpin. The anti-invasive mechanism of this compound was related to its ability to increase the expression levels of tissue inhibitor of metallopeptidase-1 (TIMP-1) and TIMP-2. Consequently, the degradation of matrix metalloproteinases (MMPs) was inhibited, and thus resulting in a decreased metastasis of the A549 cells [197].

Marthiapeptide A (**139**) was a novel cyclic peptide isolated from a marine-derived actinomycete *Marinactinospora thermotolerans*. This compound displayed significant cytotoxicity towards SF-268, MCF7, NCI-H460 (human lung tumor cells) and HepG2 cancer cells with IC_50_ values ranging from 0.38 to 0.52 μM. Its cytotoxic potency was actually 5 to 10 folds higher than the positive control compound—cisplatin. In addition to its potent cytotoxicity, marthiapeptide A was demonstrated to exert anti-microbial activity against a panel of Gram-positive bacteria [198]. A rational synthetic route of marthiapeptide A had been described to produce a sufficient amount of this compound for further bioactivity evaluation [199]. The synthesis of additional analogs may be inspired by the preparation of marthiapeptide A and its macrocyclic backbone.

Another cytotoxic cyclic peptide named mechercharstatin A (former name, mechercharmycin A) (**140**) was isolated from the marine-derived *Thermoactinomyces* sp. YM3-251. This compound exhibited strong cytotoxicity against A549 and Jurkat cells with IC_50_ values of 0.040 and 0.046 μM, respectively [200]. The GI_50_ values of mechercharstatin A against A549, HT29 and MDA-MB-231 cells were determined to be 0.03, 0.04 and 0.09 μM, respectively. The anti-proliferative action of mechercharstatin A was associated with cell cycle arrest at the G2 phase and apoptotic events [201]. Although the chemical structure of mechercharstatin A is highly similar to telomestatin (**119**), a telomerase inhibitor described previously in this section, no significant inhibition on telomerase, DNA polymerase or reverse transcriptase was observed after the treatment with mechercharstatin A [198]. Therefore, more specific and concrete pharmacological studies are needed in order to elucidate its special mechanism of action.

Urukthapelstatin A (**141**) was another a cytotoxic cyclic peptide purified from the cultured mycelia of the marine-derived *Thermoactinomycetaceae* bacterium *Mechercharimyces asporophorigenens* YM11-542. This compound was assayed to be cytotoxic against a spectrum of cancer cell lines with the mean GI_50_ value of 1.55 μM, among which lung and ovarian cancer cell lines were the most sensitive. This peptide was structurally similar to telomestatin (**119**) as well; however, urukthapelstatin A showed no significant effect on telomerase inhibition. Thus, the cytotoxic activity of this compound is derived a mechanism very different from telomestatin, and its actions are more or less like the ones of mechercharstatin A (**140**) [202]. Regarding the synthetic aspect, both total synthesis and biomimmetic synthesis of urukthapelstatin A have been achieved correspondingly, which allow an efficient production of new analogs and further extensive SAR studies [203,204,205].

### 4.3. Anticancer Cyclopeptides Derived from Cyanobacteria

Cyanobacteria are prokaryotes with ability to produce oxygen through photosynthesis [206]. There are around 150 genera and 2000 species in the phylum cyanobacteria. A great number of peptides found in cyanobacteria have been confirmed to be cytotoxic against human cancer cells, and some of them were used as the structural scaffolds for the synthesis of novel anticancer agents [207]. The chemical structures of these cyclopeptides are provided in Figure 8 and their activities were listed in Table 3.

Microcystins are a kind of cyanopeptides being the most extensively studied due to their severe toxicological influence in the ecosystem [208]. Accumulating evidence showed that such toxins can cause human poisoning because they widely exist in drinking water even after filtration [209]. Microcystins are a category of toxins with a cyclic heptapeptide structure produced by some cyanobacterial blooms. Over 80 microcystins have been isolated so far [210]. Among them, microcystin-LR (**142**) is the most common encountered variant owing to its potent toxicity. Microcystin-LR is a cyclic heptapeptide mainly produced by *Microcystis aeruginosa* [211]. It has been broadly reported that micrcystin-LR is a significant pathological agent causing carcinogenesis and organ injury [212]. When human fetal L-02 hepatocytes were chronically exposed to low-dose microcystin-LR, increased tumor formation was observed as a result of gankyrin activation [213]. Moreover, microcystin-LR has been suggested with great potential for immunotoxicity and may lead to various diseases and even cancers [211]. However, microcystin-LR has also been reported with notable cytotoxicity against a wide spectrum of cancer cells. Particularly, microcystin-LR was demonstrated to directly interact with organic-anion-transporting polypeptides (OATP1B1 and 1B3), and consequently resulted in remarkable anti-proliferative and pro-apoptotic effects in pancreatic cancer cells. The IC_50_ values of microcystin-LR against both human pancreatic cancer BxPC-3 and MIA PaCa2 cells were determined to be 83.50 nM and 2.14 µM, respectively [214]. Niedermeyer et al. found that OATP1B1 and 1B3 are the essential membrane transporters for microcystin to be uptaken into human cells. When compared to the normal HeLa cells, microcystin-LR was more potent in inhibiting the proliferation of HeLa cells overexpressing OATP1B1, with an IC_50_ value of around 1 nM. Thus, microcystin-LR is of great potential to be an anticancer agent against OATP1B1 transporter-overexpressing tumor cells [215]. Furthermore, OATP1B3 transporters-overexpressing cancer cells were found particularly sensitive to a series of microcystin analogs [216]. Taken together, microcystin-LR, as well as its analogs are worthy for further studies on their anticancer mechanisms.

Apratoxins are a novel family of cytotoxic cyclodepsipeptides, and they were isolated from cyanobacterium *Lyngbya* sp. with potent anti-proliferative effects. Apratoxin A (**143**) was a natural peptide possessing a signature cyclic structure purified from the secondary metabolites of marine cyanobacterium *Lyngbya majuscula*. Apratoxin A showed significant in vitro cytotoxicity against KB and LoVo cancer cells with IC_50_ values of 0.52 and 0.36 nM, respectively. Nevertheless, this antitumor cyclopeptide was toxic and poorly tolerated in xenograft mouse models [217]. Mechanistic studies have become the core researching area of this compound. The in vitro anti-proliferative activity of Apratoxin A was associated with the G1 cell cycle arrest and apoptosis via an inhibition of signal transducer and activator of transcription 3 (STAT3) [218]. Liu et al. [219] discovered that apratoxin A could induce degradation of cancer-related receptors, tyrosine kinases, by inhibiting their *N*-glycosylation and cotranslational translocation in the secretory pathway. A subsequent biochemical study demonstrated that apratoxin A prevented the protein cotranslational translocation by a direct blockade of the protein translocation channel Sec61α [220]. Further, Huang et al. [221] found that the in vivo toxicity of apratoxin A was the result of severe pancreatic atrophy upon the high-level exposure of the drug. They also identified Sec61 complex was a molecular target of apratoxin A in the secretory pathway. Taken together, the cytotoxicity mechanisms of apratoxin A are complicated, and its in vivo application remains controversial. However, apratoxin A is an inspiring tool to the study of the secretory pathway and cotranslational translocation inhibition for the development of anticancer drugs. Two other members of the family, apratoxins B (**144**) and C (**145**) are also derived from the secondary metabolites of *Lyngbya* sp. These two cyclopeptides also delivered cytotoxicities against human cancer cell lines, KB (IC_50_ = 21.3 and 1.0 nM, respectively) and LoVo cells (IC_50_ = 10.8 and 0.73 nM, respectively). The results indicated that apratoxin C was less potent than apratoxins A and B, though apratoxin C is different from apratoxin B by only the position of a methyl group [222]. Likewise, isolated from the marine cyanobacteria *L. majuscula* and *L. sordida*, apratoxin D (**146**) has a similar structure to apratoxin A. It exerted potent cytotoxicity to cancer cells as well. Its IC_50_ value against H460 cells was determined to be 2.6 nM. From the SAR assessments of apratoxins A-D, we noticed that the additional lipopeptide tail was responsible for the cytotoxic activities of this type of compounds; however, Suyama et al. made a contrary claim [223]. This SAR information is extremely useful to the design of additional analogs. Apratoxin E (**147**) was another congener isolated from *L. bouillonii* strain PS372. However, comparing to the IC_50_ values of apratoxin A (IC_50_ = 1.4, 10 and 10 nM, respectively), apratoxin E (IC_50_ = 21, 72 and 59 nM, respectively) showed less potency when testing against the HT29, HeLa and U2OS (human bone osteosarcoma epithelial) cells. The cell viability test results indicated that the loss of a hydroxy group and the formation of a conjugated double bond in apratoxin E attributed to the significant reduction of bioactivities [224]. Apratoxins F (**148**) and G (**149**) were two novel cytotoxins obtained from *L. bouillonii* cyanobacterium collected in the Central Pacific. When incubated with H460 cancer cells, the inhibitory IC_50_ values of apratoxins F and G were found to be 2 and 14 nM, respectively; however, they are less cytotoxic to HCT116 cells [apratoxins F (IC_50_ = 36.7 nM) and G (insufficient to test the IC_50_ value)]. According to the SAR assessment results, the *N*-methyl alanine residue was not the key functional group for the cytotoxicity of apratoxins [225]. Overall, disparate moieties and regions are usually crucial for biological activities, which can be of great use for further synthesis and drug development.

Unlike aurilide (**66**) isolated from sea hare, aurilides B (**150**) and C (**151**) were obtained from cyanobacterium *L. majuscula*. Both aurilides B and C were cyclic depsipeptides with potent in vitro cytotoxic activities. In H460 cells and mouse neuroblastoma (neuro-2a) cells, the LC_50_ values of aurilide B were found to be 0.04 and 0.01 µM, respectively whereas the LC_50_ values of aurilide C were found to be 0.13 and 0.05 µM, respectively. Moreover, aurilide B was demonstrated to induce microfilament network disruption against A-10 smooth muscle cells with an IC_50_ value of 2.9 µM. In an in vivo experiment, aurilide B was shown to deliver net tumor cell killing activity in the NCI’s hollow fiber assay [226]. A quantitative shRNA screening experiment illustrated that cytotoxicity of aurilide B was derived from mitochondrial apoptosis, which was tightly regulated by the gene sodium/potassium-transporting ATPase subunit alpha-1 (ATP1A1) [227]. Collectively, the promising bioassay data suggested that aurilide B has potential for further preclinical studies as a novel anticancer agent.

Lagunamides A-C (**152**–**154**) were cyclic depsipeptides isolated from *L. majuscula* with anti-proliferative effects against several cancer cell lines [228]. Tripathi et al. assessed the inhibitory activities of lagunamides A against the cancer cell lines P388, A549, PC3, HCT8 (human colorectal adenocarcinoma) as well as the SK-OV3 cells. The IC_50_ values of lagunamides A were found ranging from 1.64 to 6.4 nM. Lagunamide B was only tested for its cytotoxicities against P388 and HCT8 cell lines; its IC_50_ values were determined to be 20.5 and 5.2 nM, respectively. Subsequent studies revealed that the cytotoxic effects of lagunamides A and B was highly associated with the induction of mitochondrial apoptosis [229]. As for lagunamide C, its IC_50_ values varied from 2.1 nM to 24.2 nM against a panel of cancer cell lines such as P388, A549, PC3, HCT8 and SK-OV3 [230].

Another cyclic depsipeptide identified from *L. majuscula* was homodolastatin 16 (**155**), which was a homologue of the potent anticancer agent dolastatin 16. Compared to dolastatin 16, homodolastatin 16 only exhibited moderate activities against esophageal (WHCO1 and WHCO6) and cervical (ME180) cancer cell lines with IC_50_ values of 4.19, 9.85 and 8.09 µM, respectively [231].

Lyngbyabellin A (**156**) was another cytotoxic cyclic depsipeptide derived from Guamanian strain of *L. majuscula*. Other than cytotoxic activities against KB cells (IC_50_ = 0.04 µM) and LoVo cells (IC_50_ = 0.72 µM), lyngbyabellin A also exhibited specific cytoskeletal-disrupting effects to microfilaments in A-10 cells at concentrations ranging from 0.01 to 7.24 µM [232]. To further investigate the properties of this natural cyanobacterium metabolite, the total synthesis of lyngbyabellin A was subsequently achieved [233]. Similar to lyngbyabellin A, lyngbyabellins D (**157**), F (**158**) and H (**159**) were also isolated from *L. majuscula*. They exhibited cytotoxicity against H460 and KB cancer cells with IC_50_ or LC_50_ values ranging from 0.1 to 0.4 µM [234]. In the subsequent studies, Choi et al. identified lyngbyabellin N (**160**) from *Moorea bouillonii*, a rather new cytotoxic member of the lyngbyabellin class. This compound showed a potent anticancer potential against the CRC HCT116 cells with IC_50_ value of 40.9 nM [235]. Although a series of lyngbyabellins had been proved with anticancer properties, limited studies on these congeners largely restricted the ultimate exploration of these compounds as potential therapeutic agents.

Palmyramide A (**161**), another cyclic depsipeptide, was first isolated from *L. majuscula*. In the H460 cell line, palmyramide A displayed merely a weak cytotoxicity (IC_50_ = 39.7 µM). However, in neuro-2a cells, palmyramide A led to the blockage of the voltage-gated sodium channel with an IC_50_ value of 17.2 µM [236]. As a result, significant bioactivity improvements and further mechanistic studies of palmyramide A are needed.

Hoiamides A (**162**) and B (**163**) were two cyclic depsipeptides isolated from two distinct collections of marine cyanobacteria obtained from Papua New Guinea. Hoiamide A was a potent bioactive cyclic depsipeptide found in the assemblage of *Lyngbya majuscula* and *Phormidium gracile* [237], while hoiamide B was extracted from a sample of marine cyanobacteria collected in Gallows reef, the species of which was still unclear [238]. In a previous report, hoiamide A was shown to strongly inhibit voltage-gated sodium channels so that to activate the Na^+^ influx in mouse neocortical neurons [237]. In addition, Cao et al. found that in neocortical neurons, hoiamide A induced neuronal death through activating the c-Jun *N*-terminal kinase (JNK) and caspase signaling pathways [239]. In the aspect of their anticancer effects, both hoiamides A and B showed cytotoxicity against H460 cells with IC_50_ values of 11.2 and 8.3 µM, respectively [238]. Collectively, hoiamides A and B possess only moderate anticancer properties, which are still insufficient for further drug promotion.

Wewakazole (**164**) was a cyclic dodecapeptide derived from L. majuscula in Papua New Guinea [240]. A mass spectrometry-guided isolation afforded the isolation of wewakazole B (**165**), another novel cyclic peptide congener of wewakazole, from cyanobacterium Moorea peoducens. As cyanobactins, the most prominent bioactivity of the two compounds is cytotoxicity. When testing against H460 cells, wewakazole B (IC_50_ = 1.0 µM) was more cytotoxic than wewakazole (IC_50_ = 10 µM). Besides, wewakazole B also exhibited cytotoxicity against MCF7 cells with an IC_50_ value of 0.58 µM. Different from wewakazole, wewakazole B was inactive in siderophore assay [241]. The diverging bioactivities implicated that the iron-binding ability in their structures is crucial and deserves a detailed investigation. In recent years, both wewakazole and wewakazole B have been totally synthesized. With the availability of such synthetic strategy, steady and sufficient supply of these cyanobactins can be warranted for further biological evaluations [242,243].

Hectochlorin (**166**) was isolated from *L. majuscula*. It is a cyclic lipopeptide with moderate cytotoxicity against several cancer cell lines. Hectochlorin showed an average GI_50_ value of 5.1 µM in the NCI-60 screening assay; the colon, melanoma, ovarian and renal cell lines were the most sensitive to this compound [244]. In another bioassay, hectochlorin displayed cytotoxicity against KB and NCI-H187 (human small cell lung cancer) cells with ED_50_ values of 0.86 and 1.20 µM, respectively [245]. Further, this cyclic lipopeptide was able to induce actin polymerization and affect cell cycle regulation in those highly dividing cells. Apart from its cytotoxicity to cancer cells, hectochlorin was also found to be biologically active to yeasts and fungi [244]. Total synthesis and biosynthesis at the genetic level have been extensively studied in order to secure an abundant supply of this compound for further chemical and biological investigations [246,247].

Desmethoxymajusculamide C (DMMC) (**167**), a cyclic depsipeptide found in the process of a cytotoxicity-guided fractionation, was isolated from *L. majuscula*. The cyclic DMMC exhibited potent selectivity and cytotoxicity against the cancer cell lines HCT116, H460 and MDA-MB-435 with IC_50_ values of 0.02, 0.063 and 0.22 µM, respectively. Furthermore, a ring-opened form of DMMC was produced by means of base hydrolysis. Remarkably, the linear form of DMMC shared the same potency of cytotoxicity against the same panel of cancer cell lines (IC_50_ = 0.016, 0.094 and 0.23 µM, respectively). When acting against A10 cells, both cyclic and linear DMMCs disrupted the cellular microfilament networks and caused dramatic alterations in cell morphology, which were shown as the result of cellular apoptosis. Overall, these biological data indicated the potential anticancer efficacy of both cyclic and linear forms of DMMC [248]. The significant antitumor activities of DMMC are in need of further investigations, such as SAR and in vivo studies.

Hantupeptin A (**168**), another cyclodepsipeptide isolated from *L. majuscula*, was first discovered in Singapore. The compound displayed in vitro cytotoxicities against human lymphoblastic leukemia (MOLT-4) and MCF7 cells with IC_50_ values of 32 nM and 4.0 µM, respectively [249]. A subsequent bioassay-guided investigation of the same cyanobacterium led to purification of two other novel cyclic peptides, hantupeptins B (**169**) and C (**170**). Both compounds exhibited cytotoxicities when tested against MOLT-4 (IC_50_ = 0.2 and 3.0 µM, respectively) and MCF7 (IC_50_ = 0.5 and 1.0 µM, respectively) cells [250]. Furthermore, hantupeptins A-C also showed 100% brine shrimp toxicity when tested at 10 and 100 ppm [249,250].

Two cyclic peptides isolated from *L. majuscula* were designated as laxaphycins A (**171**) and B (**172**). Laxaphycin B showed significant cytotoxicities against the CCRF-CEM drug-sensitive human leukemic lymphoblasts (IC_50_ = 1.11 µM), as well as the vinblastine-resistant subline overexpressing Pgp CEM/VLB_100_ (IC_50_ = 1.02 µM) and CEM/VM-1 subline with DNA topoisomerase II alteration (IC_50_ = 1.37 µM). However, laxaphycin A was found to be inactive towards these cell lines at 20 µM [251]. Subsequently, Bonnard et al. demonstrated the anti-proliferative effect of laxaphycin B on several solid tumor cancer cell lines, including A549, MCF7, PA1 (human ovarian teratocarcinoma) and PC3, with IC_50_ values ranging from 0.19 to 1.0 µM. Furthermore, obvious biological synergism was observed on DLD1 (human colorectal adenocarcinoma) and MDR leukemic cells in the presence of laxaphycins A and B. Similar to the compounds mentioned previously, laxaphycins B2 (**173**) and B3 (**174**) were also isolated from the same assemblage of the cyanobacteria. These two novel cyclic peptides, displayed merely moderate cytotoxicities against the three leukemic cell lines above [252]. Laxaphycin B4 (**175**) was a new member of B-type laxaphycins, and it was discovered from the marine cyanobacterium *Hormothamnion enteromorphoides*. This compound yielded growth-inhibitory activity against HCT116 (IC_50_ = 1.7 µM); synergistic effect was obtained when it was used in combination with laxaphycin A [253].

Obyanamide (**176**) was a cytotoxic cyclic depsipeptide isolated in *L. confervoides*. From the MTT assay results, we noticed that this compound exhibited cytotoxicity against KB and LoVo cells with IC_50_ values of 0.97 and 5.24 µM, respectively [254]. To further promote biochemical utilization of obyanamide, several analogs were synthesized for biological evaluation, among which the cyclic analog 9b possessed the most potent activity. According to the SAR assessments, *β*-amino acid was an essential moiety for the bioactivities of obyanamide and its analogs [255]. Compared to the nature-derived obyanamide, its analogs are of greater significance for further biological studies.

Grassypeptolide A (**177**) was a macrocyclic depsipeptide with anticancer potential isolated from *L. confervoides*. Anti-proliferative activities of grasssypeptolide A were investigated in the U2OS (IC_50_ = 2.2 µM), HeLa (IC_50_ = 1.0 µM), HT29 (IC_50_ = 1.5 µM) and IMR-32 (IC_50_ = 4.2µM) cancer cell lines [256]. Subsequently, another study on the extract of the same cyanobacterium led to the purification of grassypeptolides A-C (**177**–**179**). The in vitro anti-proliferative effects of these compounds were evaluated in HT29 (IC_50_ = 1.22 µM, 2.97 µM and 76.7 nM, respectively) and HeLa (IC_50_ = 1.01 µM, 2.93 µM and 44.6 nM, respectively) cells. According to the cell viability assay results, grassypeptolide C was 16-23 times more potent than grassypeptolide A. Moreover, both grasspeptolides A and C could induce cell cycle arrest at the G1 phase at relatively low concentrations (10 µM and 316 nM, respectively); they could also induce the G2/M phase arrest at higher concentrations (31.6 µM and 1 µM, respectively). These two compounds have the capability to bind to metal ions such as Cu^2+^ and Zn^2+^, which may denote a partial mechanism of their cytotoxicities [257]. The study Liu et al. revealed that grassypeptolide A increased the cleavage of PARP and decreased the levels of bcl-2 and bcl-xL. As for the induction of cell cycle arrest, it downregulated cyclin D, but upregulated P27 and P21 [258]. Grassypeptolides D (**180**) and E (**181**), two novel members of the family, were isolated from the marine cyanobacterium *Leptolyngbya* sp. These two compounds were cytotoxic to HeLa cells (IC_50_ = 335 and 192 nM, respectively) and mouse neuro-2a blastoma cells (IC_50_ = 599 and 407 nM, respectively) [259]. To promote these grassypeptolides for further drug development, total synthesis and mechanistic studies are desperately needed.

Palauamide (**182**), a cyclic depsipeptide extracted from a bioassay-guided fractionation, was confirmed as a potent cytotoxin isolated from *Lyngbya* sp. in Palau. When cultured with KB cells, palauamide yielded significant cytotoxicity with an IC_50_ value of 13 nM [260]. Palauamide was totally synthesized, along with its diastereomers and other analogs, which also showed potent cytotoxicities [261,262]. However, more analogs are in need to be synthesized for further elucidation of their biological potential and SAR studies.

Coibamide A (**183**) was a cyclic depsipeptide that was first isolated from the marine filamentous cyanobacterium *Leptolyngbya* sp. in Coiba National Park, Panama. The preliminary biological activity screening showed that this compound displayed potent cytotoxicity against NCI-H460 and mouse neuro-2a cells with LC_50_ < 17.9 nM. The evaluation against the NCI’s in vitro 60 cancer cell line panel highlighted its remarkable anti-proliferative response. From the flow cytometric analysis, coibamide A was shown to induce cell cycle arrest at the G1 phase [263]. Subsequently, Hau et al. [264] demonstrated that coibamide A induced mammalian target of rapamycin (mTOR)-independent autophagy and cell death in human glioblastoma cells. In the glioblastoma xenograft models, the expression levels of vascular epithelial growth factor A/vascular epithelial growth factor receptor 2 (VEGFA/VEGFR2) were markedly suppressed upon the treatment with coibamide A, so that the size of tumor xenografts was largely reduced from 2000 mm^3^ to 200-300 mm^3^. However, different degrees of weight loss were observed in the animal experiment. Such result implicated that targeted delivery and toxicity profile should be improved for drug development [265].

Ulongamides A-E (**184**–**188**) were five β-amino acid-containing cyclopeptides derived from the marine cyanobacterium *Lyngbya* sp. in Palau. These ulongamides exhibited moderate in vitro cytotoxicity against KB and LoVo cells with IC_50_ values around 1 and 5 µM [266]. The total synthesis of ulongamide A as well as its two congeners (ulongmides B and C) had been previously accomplished [267,268]. In the long run, structural optimization is needed to improve the bioactivity potency of this type of compounds.

Ulongapeptin (**189**), another cytotoxic cyclopeptide, was isolated from *Lyngbya* sp. in Palau. This compound yielded cytotoxicity against KB cells with an IC_50_ value of 0.63 µM [269]. No further synthetic or biological approaches of this compound have been reported thereafter.

The marine cyanobacterium *Okeania* sp. is the origin of the cytotoxic cyclodepsipeptide odoamide (**190**). As another new member of aurilide class, odoamide has been reported with a significant cytotoxicity against HeLa S3 cells (IC_50_ = 26.3 nM) and a moderate toxicity to brine shrimp [270]. From the SAR assessment of synthetic odoamide isomers, the in vitro cytotoxicity was mostly affected by the serum protein binding of odoamide derivatives. Nevertheless, the macrocyclic structure of odoamide and derivatives is not crucial to the alteration of membrane permeability [271].

Cryptophycin 1 (**191**), a cyclic depsipeptide isolated from cyanobacterium *Nostoc* sp., was first discovered as an antifungal agent two decades ago. Subsequent studies indicated that cryptophycin 1 was also an anti-proliferative agent with significant antitumor activities [272,273]. A great number of experiments showed that cryptophycin 1 yielded inhibitory activities against various cancer cell lines such as L1210 (IC_50_ = 4 pM), SKOV3 (IC_50_ = 7 pM) and MCF7 (IC_50_ = 16 pM). Some recent reports demonstrated that cryptophycin 1 provided an anticancer potency ~100-fold higher than the mainstay anticancer drug paclitaxel; therefore, a great interest was immediately prompted to investigate the mechanism of action of this compound [274,275]. According to the study of Kerksiek et al., cryptophycin 1 was an effective inhibitor of tubulin polymerization, by which microtubules were depolymerized into linear polymers [276]. When exposed to stoichiometric amounts of proteins, cryptophycin 1 elicited extensive aggregates of tubulin [277]. Even at picomolar concentrations, cryptophycin 1 was found to cause cell cycle arrest at the G2/M phase via its actions on the microtubules [275]. Though inhibitory effect on macromolecular synthesis of cryptophycin 1 had been observed in cancer cells, the slight downregulation of DNA, RNA and proteins was not well correlated to its anticancer potency [278]. Up to now, several analogs of cryptophycin 1, such as cryptophycin 52, have been synthesized in order to achieve higher anticancer potency [279].

Largazole (**192**) is a cyclic depsipeptide isolated from the cyanobacterium of *Symploca* sp. This compound was found to have good selectivity against different types of cancer cells. For instance, largazole was less cytotoxicy to the nontransformed murine mammary epithelial MDA-MB-231 cells (GI_50_ = 122 nM, LC_50_ = 272 nM), but it was more susceptible to the invasive transformed human mammary epithelial (NmuMG) cells (GI_50_ = 7.7 nM, LC_50_ = 117 nM). Similarly, this cyclopeptide displayed higher inhibitory effect against U2OS transformed fibroblastic osteosarcoma (GI_50_ = 55 nM, LC_50_ = 94 nM) than NIH3T3 nontransformed fibroblasts (GI_50_ = 480 nM, LC_50_ > 8 µM) [280]. Based on the pharmacological studies carried out by Bowers et al., largazole was considered a potent inhibitor of class I HDACs [281]. In the subsequent SAR assessments of largazole and its derivatives, largazole was shown to exhibit good selectivity towards cancer cell lines HCT116 and A549 with GI_50_ values of 0.08 µM and 0.32 µM, respectively [282]. The in vitro cytotoxicity of largazole in the HCT116 cells was plausibly derived from its induction of cell cycle arrest and apoptosis. When tested in other CRC cells (e.g., HT29 and HCT15), this compound was found to induce histone H3 hyperacetylation. The further in vivo studies indicated that the anticancer effect of largazole was highly associated with an attenuation of the protein kinase B (AKT) pathway and a modulation of cell cycle regulators [283]. It is widely accepted that ubiquitination of protein is important for the vast majority of eukaryotic cellular functions; regular protein ubiquitination is halted in many cancers. Liu and his co-workers reported that largazole was able to inhibit the ubiquitin activating enzyme E1 to disturb ubiquitin conjugation with p27^Kip1^ and telomeric repeat-binding factor 1 (TRF1) in vitro [284]. In addition, largazole was demonstrated to induce the expression of osteoblast differentiation markers, such as ALP. Taken together, largazole is suggested to activate osteogenesis [285]. While showing several intriguing and multidisciplinary functions, largazole is undoubtedly a potential therapeutic agent for further development.

Studies on a cyanobacterium of *Symploca* sp. From Papua New Guinea led to the isolation of a novel cyclopeptide—symplocamide A (**193**). This compound showed potent cytotoxicity against H460 and neuro-2a cells with IC_50_ values of 40 nM and 29 nM, respectively. In fact, symplocamide A shares a structure very similar to protease inhibitors; thus, it can be served as a potent chymotrypsin inhibitor (IC_50_ = 0.38 µM). The role of protease inhibition in cancer therapy is worth to be studied. On the other hand, symplocamide A was reported to show moderate anti-parasite activities against malaria, Chagas disease and leishmaniasis [286]. Solid phase total synthesis of symplocamide A had been accomplished by Kaiser and his co-workers, which has laid a groundwork for further studies of this type of compounds [287].

Two other cyclic depsipeptides had been isolated from *Symploca* sp. in Palau, and they were named tasipeptins A (**194**) and B (**195**). These two compounds exhibited cytotoxicities against KB cells with IC_50_ values of 0.93 and 0.82 µM, respectively [288]. Due to the presence of Ahp, a symbolic structural moiety of cyanobacterial metabolites, tasipeptins A and B were considered as potential protease inhibitors. The total synthesis of these two compounds had been completed by Kaiser and his co-workers [289]. It is believed that ever-growing investigations on Ahp-containing cyclic depsipeptides can make good contributions to the further rational design of novel anticancer lead compounds.

Cytotoxicity-directed fractionation of cyanobacterium *Symploca* cf. *hydnoides* from Guam led to the isolation of a series of cyclic depsipeptides, veraguamides A-G (**196**–**202**). The results of several cell viability assays showed that veraguamides A-G displayed weak to moderate cytotoxicities against HT29 and HeLa cells with IC_50_ values varying from 0.84 to 49 µM and 0.54 to 49 µM, respectively. From the SAR assessments, the seven semi-synthetic analogs of veraguamide G explicitly denoted the sensitive positions of veraguamides that were responsible for their cytotoxic activities [290].

Floridamide (**203**), a novel cyclic depsipeptide containing four amino acid units, was discovered in a bioassay-guided investigation of *Moorea producens*, a species of filamentous cyanobacteria collected from Florida, USA. This compound displayed cytotoxic activities against H460 and mouse neuro-2a neuroblastoma cells with the EC_50_ values of around 1.89 × 10^−5^ µM [291].

Viequeamide A (**204**) was a highly toxic cyclic depsipeptide found in the cyanobacterium *Rivularia* sp. near the island of Vieques. It was found to be strongly cytotoxic against H460 cells with an IC_50_ value of 60 nM. As a member of the “kulolide superfamily”, the biological properties of viequeamied A were reported to be highly divergent [292]. The total synthesis of this compound had been already achieved, so that the steady and sufficient supply should facilitate further intriguing studies on SAR assessment and mechanism of actions of this type of compounds [293].

**Table 3 ijms-22-03973-t003:** Bacteria-derived anticancer cyclopeptides.

Name	Biological Source	Anticancer Activity	Reference
Anticancer Cyclopeptides Derived from Non-Marine Bacteria			
YM753 (spiruchostatin A/OBP-801) (**118**)	*Pseudomonas* sp.	In vitro growth inhibitory ceffects on many cancer cells including WiDr, melanoma (GI_50_ = 5.0 nM), Calu-3 (GI_50_ = 1.6 nM), HOS (GI_50_ = 3.8 nM) cells and etc.; In vivo antitumor activity on WiDr xenograft model.	[168,169,170,171]
Telomestatin (**119**)	*Streptomyces anulatus* 3533-SV4	Cytotoxicity against MiaPaCa (IC_50_ = 0.5 µM) cell and four neuroblastoma cells (IC_50_ = 0.8 to 4.0 µM). Specific telomerase inhibitor in cancer cells inhibition (inhibit cellular senescence).	[172,173,174,175,176,177,178]
Anticancer Cyclopeptides Derived from Marine Bacteria			
Thiocoraline (**120**)	Gram-positive bacteria *Micromonospora* sp. ACM2-092 and *Micromonospora* sp. ML1	Cytotoxicity against P388 (IC_50_ = 1.72 nM), A549 (IC_50_ = 1.72 nM), MEL28 (IC_50_ = 1.72 nM) and HT29 (IC_50_ = 8.64 nM) and MTC-TT (IC_50_ = 7.6 nM) cells.	[181,182,183,184,185]
Ohmyungsamycins A (**121**) and B (**122**)	*Streptomyces* sp.	Cytotoxicity against HCT116, A549, SNU-638, MDA-MB-231 and SK-HEP-1 cells. (IC_50_ = 359 to 816 nM and 12.4 to 16.8 µM, respectively)	[186,187]
Arenamides A (**123**) and B (**124**)	*Salinispora arenicola*	Cytotoxicity against HCT116 (IC_50_ = 1.97 and 2.99 μM, respectively) cell.	[188,189]
Piperazimycins A-C (**125**–**127**)	*Streptomyces* sp.	Cytotoxicity against HCT116 cell. (Average GI_50_ = 0.1 µM) Piperazimycin A showed a mean GI_50_ of 100 nM in NCI 60 cancer cell line panel screening.	[190,191]
Kailuins A-D (**128**–**131**)	Gram-negative bacterium BH-107	Inhibitory activities against A549, MCF7 and HT29 cells. (IC_50_ = 2.66 to 5.52 µM)	[192]
Mixirins A-C (**132**–**134**)	*Bacillus* genus	Anti-proliferative activity of HCT116 (IC_50_ = 0.65, 1.6 and 1.26 μM, respectively) cell.	[193]
Bacilohydrin A (**135**)	*Bacillus* sp. SY27F	Cytotoxicity against HepG2, MCF7 and DU145 cells. (IC_50_ = 50.3 to 175.1 nM)	[195]
Dolyemycins A (**136**) and B (**137**)	*Streptomyces griseus* subsp. *griseus* HYS31	Anti-proliferative activities against A549 cells. (IC_50_ = 1 and 1.2 µM, respectively)	[196]
Sungsanpin (**138**)	*Streptomyces* sp.	Anti-invasion effects on A549 cell.	[197]
Marthiapeptide A (**139**)	Actinomycete *Marinactinospora thermotolerans*	Cytotoxicity against SF-268, MCF7, NCI-H460 and HepG2 cell. (IC_50_ = 0.38 to 0.52 µM)	[198,199]
Mechercharstatin A (**140**)	*Thermoactinomyces* sp. YM3-251	Cytotoxicity against Jurkat cell (IC_50_ = 4.6 × 10^−8^ M), A549 (IC_50_ = 0.03 µM), HT29 (IC_50_ = 0.04 µM) and MDA-MB-231 (IC_50_ = 0.09 µM) cells.	[198,200,201]
Urukthapelstatin A (**141**)	*Mechercharimyces asporophorigenens* YM11-542	Cytotoxicity against several cancer cell lines. (Mean GI_50_ = 1.55 μM)	[202,203,204,205]
Anticancer Cyclopeptides Derived from Cyanobacteria			
Microcystin-LR (**142**)	Mainly from *Microcystis aeruginosa*	Cytotoxicity against BxPC-3 (IC_50_ = 83.50 nM) and MIA PaCa2 (IC_50_ = 2.14 µM) and transfected HeLa (IC_50_ = 1 nM) cells.	[211,212,213,214,215,216]
Apratoxin A (**143**)	*Lyngbya* sp.	Cytotoxicity against KB (IC_50_ = 0.52 nM), LoVo (IC_50_ = 0.36 nM) and HCT116 (IC_50_ = 1.21 nM) cells.	[217,218,219,220,221]
Apratoxins B (**144**) and C (**145**)	*Lyngbya* sp.	Cytotoxicity against KB (IC_50_ = 21.3 and 1.0 nM, respectively) and LoVo (IC_50_= 10.8 and 0.73 nM, respectively) cells.	[222]
Apratoxin D (**146**)	*Lyngbya majuscula* and *Lyngbya sordida*	Cytotoxicity against H460 (IC_50_ = 2.6 nM) cell.	[223]
Apratoxin E (**147**)	*Lyngbya bouillonii* strain PS372	Cytotoxicity against HT29 (IC_50_ = 21 nM), HeLa (IC_50_ = 72 nM) and U2OS (IC_50_ = 59 nM) cells.	[224]
Apratoxins F (**148**) and G (**149**)	*Lyngbya bouillonii*	Cytotoxicity against H460 (IC_50_ = 2 and 14 nM, respectively) and HCT116 cells.	[225]
Aurilide B (**150**)	*Lyngbya majuscula*	In vitro cytotoxicity against H460 (LC_50_ = 0.04 µM) and neuro-2a (LC_50_ = 0.01 µM) cells. In vivo net tumor cell killing activity	[226,227]
Aurilide C (**151**)	*Lyngbya majuscula*	Cytotoxicity against H460 (LC_50_ = 0.13 µM) and neuro-2a (LC_50_ = 0.05 µM) cells.	[226]
Lagunamide A (**152**)	*Lyngbya majuscula*	Cytotoxicity against P388, A549, PC3, HCT8 and SK-OV3 cells. (IC_50_ = 1.64 to 6.4 nM)	[228,229]
Lagunamide B (**153**)	*Lyngbya majuscula*	Cytotoxicity against P388 (IC_50_ = 20.5 nM) and HCT8 (IC_50_ = 5.2 nM) cells.	[228,229]
Lagunamide C (**154**)	*Lyngbya majuscula*	Cytotoxicity against P388, A549, PC3, HCT8 and SK-OV3 cells (IC_50_ = 2.1 to 24.2 nM).	[228,230]
Homodolastatin 16 (**155**)	*Lyngbya majuscula*	Cytotoxicity against WHCO1 (IC_50_ = 4.19 µM), WHCO6 (IC_50_ = 9.85 µM) and ME180 (IC_50_ = 8.09 µM) cells.	[231]
Lyngbyabellin A (**156**)	*Lyngbya majuscula*	Cytotoxicity against KB (IC_50_ = 0.04 µM) and LoVo (IC_50_ = 0.72 µM) cells. Specific cytoskeletal-disrupting effects to microfilaments in A-10 cells.	[232,233]
Lyngbyabellins D (**157**), F (**158**) and H (**159**)	*Lyngbya majuscula*	Cytotoxicity against H460 and KB cells. (IC_50_ or LC_50_ = 0.1 to 0.4 µM)	[234]
Lyngbyabellin N (**160**)	*Moorea bouillonii*	Cytotoxicity against HCT116 (IC_50_ = 40.9 nM) cell.	[235]
Palmyramide A (**161**)	*Lyngbya majuscula*	Modest cytotoxicity against H460 cell (IC_50_ = 39.7 µM). Blockage of the voltage-gated sodium channel in neuro-2a cell.	[236]
Hoiamide A (**162**)	*Lyngbya majuscula* and *phormidium gracile*	Cytotoxicity against H460 (IC_50_ = 11.2 µM) cell.	[237,238,239]
Hoiamide B (**163**)	A sample of marine cyanobacteria collected in Gallows reef, unclear species	Cytotoxicity against H460 (IC_50_ = 8.3 µM) cell.	[237,238]
Wewakazole (**164**)	*Lyngbya majuscula*	Cytotoxicity against H460 (IC_50_ = 10 µM) cell.	[240,241,242,243]
Wewakazole B (**165**)	*Moorea peoducens*	Cytotoxicity against H460 (IC_50_ = 1.0 µM) and MCF7 (IC_50_ = 0.58 µM) cells.	[241,242,243]
Hectochlorin (**166**)	*Lyngbya majuscula*	Average GI_50_ value of 5.1 µM in the tests against a panel of NCI 60 tumor cell lines. Cytotoxicity against KB (ED_50_ = 0.86 µM) and NCI-H187 (ED_50_ = 1.20 µM) cells.	[244,245,246,247]
Desmethoxymajusculamide C (DMMC) (**167**)	*Lyngbya majuscula*	Selectivity and cytotoxicity against HCT116 (IC_50_ = 0.016 µM), H460 (IC_50_ = 0.094 µM) and MDA-MB-435 (IC_50_ = 0.23 µM) cells.	[248]
Hantupeptin A (**168**)	*Lyngbya majuscula*	Cytotoxicity against MOLT-4 (IC_50_ = 32 nM) and MCF7 (IC_50_ = 4.0 µM) cells.	[249]
Hantupeptins B (**169**) and C (**170**)	*Lyngbya majuscula*	Cytotoxicity against MOLT-4 (IC_50_ = 0.2 and 3.0 µM, respectively) and MCF7 (IC_50_ = 0.5 and 1.0 µM, respectively) cells.	[249,250]
Laxaphycin A (**171**)	*Lyngbya majuscula*	Obvious biological synergism when acting against DLD1 and MDR cells in the presence of laxaphycins A and B.	[251,252]
Laxaphycin B (**172**)	*Lyngbya majuscula*	Cytotoxicity against CCRF-CEM (IC_50_ = 1.11 µM), CEM/VLB_100_ (IC_50_ = 1.02 µM) and CEM/VM-1 (IC_50_ = 1.37 µM) cells. Anti-proliferative effects on A549, MCF7, PA1 and PC3 cells. (IC_50_ = 0.19 to 1.0 µM) Obvious biological synergism when acting against DLD1 and MDR cells in the presence of laxaphycins A and B.	[251,252]
Laxaphycins B2 (**173**) and B3 (**174**)	*Lyngbya majuscula*	Moderate cytotoxicities against CCRF-CEM, CEM/VLB_100_ and CEM/VM-1 cells.	[252]
Laxaphycin B4 (**175**)	*Hormothamnion enteromorphoides*	Growth-inhibitory activity and synergistic effect with laxaphycin A when acting against HCT116 (IC_50_ = 1.7 µM) cell.	[253]
Obyanamide (**176**)	*Lyngbya confervoides*	Cytotoxicity against KB (IC_50_ = 0.97 µM) and LoVo (IC_50_ = 5.24 µM) cells.	[254,255]
Grassypeptolide A (**177**)	*Lyngbya confervoides*	Cytotoxicity against U2OS (IC_50_ = 2.2 µM), HeLa (IC_50_ = 1.0 µM), HT29 (IC_50_ = 1.5 µM) and IMR-32 (IC_50_ = 4.2µM) cells.	[256,257,258]
Grassypeptolides B (**178**) and C (**179**)	*Lyngbya confervoides*	Anti-proliferative effects on HT29 (IC_50_ = 2.97 µM and 76.7 nM, respectively) and HeLa (IC_50_ = 2.93 µM and 44.6 nM, respectively) cells.	[257]
Grassypeptolides D (**180**) and E (**181**)	*Leptolyngbya* sp.	Cytotoxicity against HeLa (IC_50_ = 335 and 192 nM, respectively) and neuro-2a (IC_50_ = 599 and 407 nM, respectively) cells.	[259]
Palauamide (**182**)	*Lyngbya* sp.	Cytotoxicity against KB (IC_50_ = 13 nM) cell.	[260,261,262]
Coibamide A (**183**)	*Leptolyngbya* sp.	Cytotoxicity against NCI-H460 and neuro-2a cells. (LC_50_ < 17.9 nM)	[263,264,265]
Ulongamides A-E (**184**–**188**)	*Lyngbya* sp.	Cytotoxicity against KB (IC_50_~1 µM) and LoVo (IC_50_~5 µM) cells.	[266,267,268]
Ulongapeptin (**189**)	*Lyngbya* sp.	Cytotoxicity against KB (IC_50_ = 0.63 µM) cell.	[269]
Odoamide (**190**)	*Okeania* sp.	Cytotoxicity against HeLa S3 (IC_50_ = 26.3 nM) cell.	[270,271]
Cryptophycin 1 (**191**)	*Nostoc* sp.	Broad-spectrum cytotoxicity against cell lines such as L1210 (IC_50_ = 4 pM), SKOV3 (IC_50_ = 7 pM), MCF7 (IC_50_ = 16 pM) and etc.	[272,273,274,275,276,277,278,279]
Largazole (**192**)	*Symploca* sp.	Cytotoxicity against MDA-MB-231 (GI_50_ = 122 nM, LC_50_ = 272 nM), NmuMG (GI_50_ = 7.7 nM, LC_50_ = 117 nM), U2OS (GI_50_ = 55 nM, LC_50_ = 94 nM), NIH3T3 (GI_50_ = 480 nM, LC_50_ > 8 µM), HCT116 (GI_50_ = 0.08 µM) and A549 (GI_50_ = 0.32 µM) cells.	[280,281,282,283,284,285]
Symplocamide A (**193**)	*Symploca* sp.	Cytotoxicity against H460 (IC_50_ = 40 nM) and neuro-2a (IC_50_ = 29 nM) cells.	[286,287]
Tasipeptins A (**194**) and B (**195**)	*Symploca* sp.	Cytotoxicity against KB (IC_50_ = 0.93 and 0.82 µM, respectively) cell.	[288,289]
Veraguamides A-G (**196**–**202**)	*Symploca* cf. *hydnoides*	Cytotoxicity against HT29 (IC_50_ = 0.84 to 49 µM) and HeLa (IC_50_ = 0.54 to 49 µM) cells.	[290]
Floridamide (**203**)	*Moorea producens*	Cytotoxicity against H460 and neuro-2a cells. (EC_50_ = 1.89 × 10^−5^ µM)	[291]
Viequeamide A (**204**)	*Rivularia* sp.	Cytotoxicity against H460 (IC_50_ = 60 nM) cell.	[292,293]

## 5. Fungi-Derived Anticancer Cyclopeptides

Fungi are eukaryotic organisms with worldwide distribution and different habitats. It is estimated that biodiversity of fungi is between 2.2 to 3.8 million species around the world [294]. Moreover, fungi belong to the most prolific producers of secondary metabolites [295]. Due to their roles acting as food, medicines and pathogens, as well as their biodiversity, fungi can be abundant sources of natural therapeutic agents. Many recently isolated cyclopeptides from fungi exhibited potent anticancer activities and they will be discussed in this section. The structures of these compounds were drawn in Figure 9 and their activities were presented in Table 4.

### 5.1. Anticancer Cyclopeptides Derived from Non-Marine Fungi

1-Alaninechlamydocin (**205**) was a cyclic tetrapeptide isolated from the fungi of *Tolypocladium* sp. obtained from the Great Lakes. The compound displayed predominant anti-proliferative and cytotoxic activities toward MIA PaCa-2 cells with GI_50_ and LC_50_ values of 5.3 and 22 nM, respectively. Moreover, as a potent HDAC inhibitor, 1-alaninechlamydocin induced apoptosis and cell cycle arrest at phase G2/M as well. HDAC inhibitory activity also primarily contributed to in vitro bioactivities of the compound [296].

*Pseudoxylaria* sp. X802, a termite-associated fungus, yielded six new cyclic tetrapeptides, among which pseudoxylallemycin C (**206**) showed moderate cytotoxicity. The compound exhibited anti-proliferative effect to K562 cells with the GI_50_ value of 6.8 µM and cytotoxicity to HeLa cells with CC_50_ value of 16.68 µM. However, human umbilical vein endothelial cells (HUVEC) was also sensitive to pseudoxylallemycin C (GI_50_ = 6.96 µM), which demonstrated low specificity of the compound [297].

Cycloaspeptides F (**207**) and G (**208**) were two new members of cycloaspeptides—the ABA-containing cyclic pentapeptides discovered in different fungi. These two compounds were derived from the fungus *Isaria farinosa*, an inhabitant of *Cordyceps sinensis* obtained from Linzhi, Tibet. They showed inhibitory activities toward MCF7 cells with respective GI_50_ values of 18.7 and 15.2 µM respectively, which were comparable to the positive control 5FU (GI_50_ = 15 µM) [298].

Malformin A_1_ (MA_1_) (**209**), a cyclopentapeptide with versatile biological activities, was originally isolated from the terrestrial plant fungus *Aspergillus niger*. Liu et al. investigated its anticancer mechanism of action in human prostate cancer PC3 (IC_50_ = 130 nM) aand LNCaP (IC_50_ = 90 nM) cell lines. Coexistence of apoptosis, necrosis and autophagy were all induced by mitochondrial damage, which illustrated the mechanism of MA_1_-mediated cell death in prostate cancer cells. In addition, stimulation of the AMP-activated serine/threonine protein kinase/mammalian target-of-rapamycin (AMPK/mTOR) pathway by adding oxidative stress or decreasing ATP could lead to MA_1_-mediated autophagy [299]. In human colorectal cancer cells (SW480 and DKO1), invasive and oncogenic phenotypes were modified through p38 signaling pathway [300]. Recently, the marine-derived fungus *A. tubingenesis* also yielded MA_1_ with significant cytotoxicity against HeLa (IC_50_ = 9.48 nM) and P388 (IC_50_ = 132.9 nM) cell lines [301]. Malformin A_1_ is an early-discovered cyclic peptide with diverse biological properties and sufficient studies have demonstrated its bioactivities as well as mechanism of action. However, for further promotion to clinical stage, more supporting data in animal models shall be carried out. Another new member of the malformin class, designated as malformin E (**210**), was isolated from the endophytic fungus *A. tamarii* collected from the roots of *Ficus carica*. The compound also showed in vitro cytotoxic effects on MCF7 and A549 cells, with IC_50_ values of 0.65 and 2.42 µM, respectively. Moreover, malformin E also exhibited antimicrobial activities against several strains of bacteria and fungi [302].

Another new analog of sansalvamide A (**228**), neo-*N*-methylsansalvamide (**211**), was isolated from the fungus *Fusarium solani* KCCM90040 derived from the potato in Korea. The new cyclic pentadepsipeptide was able to inhibit in vitro cell growth of A549, SK-OV-3, SK-MEL-2 (human melanoma) and MES-SA (human uterine sarcoma) cells with EC_50_ values ranging from 10.0 to 14.74 µM [303]. Moreover, synergistic effects of neo-*N*-methylsansalvamide and paclitaxel were also observed when acting against MES-SA, HCT15 (colorectal adenocarcinoma) and two multidrug resistance sublines (MES-SA/DX5 (human uterine corpus sarcoma) and HCT15/CL05). The EC_50_ values of paclitaxel added with 3 µM neo-*N*-methylsansalvamide were significantly decreased by several orders of magnitude [304]. As a result, although the cytotoxic effect of the cyclic peptide alone was not so dominant, significant potency in reversing in vitro multidrug resistance can be promoted for further development as a promising agent with clinical function.

Cylindrocyclin A (**212**), a cyclic nonapeptide, was isolated from *Cyclindrocarpon* sp. strain A101-96 during the period of a screening of new metabolites with cytotoxic effects from fungal extracts. Six cancer cell lines were assayed for its anticancer potential, while the compound only showed moderate cytotoxicity to colorectal adenocarcinoma (COLO-320), human leukemia (HL-60), L1210 and human T lymphocyte Jurkat cell lines with IC_50_ value around 11 µM. Neither antimicrobial or nematicidal effect was observed for cylindrocyclin A [305].

Studies on the mushroom *Gymnopus fusipes* led to the isolation of two novel octadecapeptides, gymnopeptides A (**213**) and B (**214**). Both compounds are octadecacyclopeptides with natural cyclic *β* hairpins. Gymnopeptides A and B displayed significant anti-proliferative activities against HeLa, A431 (human epidermoid carcinoma), T47D, MCF7 and MDA-MB-231 cells with IC_50_ values ranging from 18.0 to 88.4 nM and 14.0 to 44.3 nM, respectively [306]. Due to their unprecedented structures and dominant growth inhibitory activity on cancer cells, to make sufficient supply for further research, total synthesis of the two compounds were successfully accomplished subsequently [307]. As a result, it is fully justified to further explore their bioactive roles and mechanism of actions.

Phomopsin F (**215**), a cyclic hexapeptide derived from the fungus *Diaporthe toxica*, was also an *N*-methylated derivative of the mycotoxin phomopsin A. The initial in vitro screening of cytotoxicity revealed its impact on viability and vitability of HepG2 cells (IC_50_ = 6.1 µM) [308].

Four cytotoxic cyclodepsipeptides, pteratides I-IV (**216**–**219**), were purified from a basidiomycete identified as *Pterula* species. Pterides I and II showed remarkable cytotoxicity against P388 cell line with IC_50_ values of 41 and 40 nM, respectively. Nevertheless, compared to the first two compounds, pteratides III and IV were much less active (IC_50_ = 7.4 and 2.9 µM, respectively) due to their much different structural discrepancy to pteratides I and II [309]. Practically, these compounds can be good candidates as anticancer agents, but their mechanism of actions as well as the underlying SAR remains to be further investigated.

### 5.2. Anticancer Cyclopeptides Derived from Marine Fungi

A cyclic tetrapeptide called asperterrestide A (**220**), with entire composition of non-protein amino acids, was first isolated from the marine-derived fungus *Aspergillus terreus* SCSGAF0162 (Figure 10). The compound exhibited cytotoxicity against U937 and MOL-4 (acute lymphoblastic leukemia) cell lines with IC_50_ values of 6.4 and 6.2 μM, respectively. Moreover, the growth of the influenza virus strains H1N1 and H3N2 could also be inhibited by asperterrestide A [310]. Just like other cyclic tetrapeptide natural products, synthesis of the compound remains challenging [311].

Cordyheptapeptides A (**221**) and B (**222**) were two bioactive cycloheptapeptides previously isolated from two different insect pathogenic fungi *Cordyceps* sp. BCC 1788 and 16173, respectively. The two cyclic peptides displayed anti-proliferative activities against KB, BC (human lymphoma), NCI-H187 and Vero (normal kidney) cells with IC_50_ values of cordyheptapeptide A measured as 0.78, 0.2, 0.18 and 14 µM, respectively, the IC_50_ values of cordyheptapeptide B as 2.0, 0.66, 3.1 and 1.6 µM, respectively. In addition, cordyheptapeptide A also exhibited moderate antimalarial activities [312]. Total synthesis of the two compounds, according to subsequent reports, had been accomplished together with other pharmacological investigations. Intriguingly, both synthetic compounds were of cytotoxic potential acting against Dalton’s lymphoma ascites (DLA) (CTC_50_ = 10.6 and 7.4 µM, respectively) and Ehrlich’s ascites carcinoma (EAC) (CTC_50_ = 14.6 and 13.56 µM, respectively) cell lines [312,313]. Accordingly, further studies could add more insights into their mechanisms of action. Chen et al. discovered two other novel congeners, Cordyheptapeptides C (**223**) and E (**224**), from a marine-derived fungus *Acremonium persicinum* in 2012. Cordyheptapeptide C displayed cytotoxicity against SF-268 and MCF7 with IC_50_ values of 3.7 and 3.0 μM, respectively. And the other one showed similar cytotoxicity against SF-268, MCF7 and NCI-H460 cell lines (IC_50_ = 3.2, 2.7 and 4.5 μM, respectively) [314]. However, no subsequent pharmacological study and in vivo experiments associated with the two compounds have been reported.

Scytalidamides A (**225**) and B (**226**) were two cyclic heptapeptides extracted from the culture broth of the marine fungus *Scytalidium* sp. from the Bahamas. When incubated with HCT116 cell line, the two compounds displayed in vitro cytotoxicity with IC_50_ values of 2.7 and 11.0 µM, respectively. Tested in the NCI 60 cell line panel, scytalidamides A and B exhibited moderate cytotoxicity with mean GI_50_ values of 7.9 and 4.1 µM, respectively. The screening test also demonstrated the most sensitive cell lines were MOLT-4 (IC_50_ = 3.0 µM) for scytalidamide A and UACC-257 (IC_50_ = 3.0 µM) for scytalidamide B [315].

Zygosporamide (**227**), a cyclic pentadepsipeptide produced by the marine-derived fungus *Zygosporium masonii*, was shown to possess potent anticancer properties. Evaluated in the NCI cell line assays (median GI_50_ = 9.1 µM), the compound yielded significant in vitro cytotoxicity against central nervous system (CNS) cancer cell line SF-268 (IC_50_ = 6.5 nM) and renal cancer cell line RXF393 (IC_50_ ≤ 5.0 nM) with high selectivity [316]. The total synthesis of zygosporamide was accomplished. However, there was a notable (about 700-fold) discrepancy of the IC_50_ values between the natural and synthetic cyclic peptides when acting against SF-268 [317].

Sansalvamide A (**228**), a natural cyclic depsipeptide, was isolated first from a marine fungus of *Fusarium* sp. living on the seagrass *Halodule wrightii*. The compound showed cytotoxicity against several cancer cells lines, including HCT116, COLO 205 and SK-MEL-2 cells with IC_50_ values of 9.7, 3.46 and 5.84 μM, respectively [318]. Sansalvamide A caused in vitro pancreatic cancer cell death by upregulation of p21 and downregulation of cyclinsD1, E, A and CDK4, which was consistent with G0/G1 cell cycle arrest [319]. Additionally, it was also identified as an inhibitor of a topoisomerase in the pathogenic poxvirus molluscum contagiosum virus (MCV) [320]. However, whether such inhibitory effect was partly related to its anticancer effect is still unclear [319]. The total synthesis of sansalvamide A had also been achieved, which laid substantial groundwork for subsequent SAR study of its derivatives [321]. Several derivatives of sansalvamide A were reported to possess much higher efficacy and selectivity against different cancer cells than the natural product [321,322,323]. Accordingly, sansalvamide A is considered as a lead compound to synthesise its congeners to improve its anticancer potential.

Clavatustides A (**229**) and B (**230**) were two novel cyclodepsipeptides derived from the fungus *Aspergillus clavatus*. *Xenograpsus testudinatus*, a kind of hydrothermal vent crab, is the habitat where they found the fungus. Jiang et al. determined that both compounds suppressed the proliferation of HCC (hepatocellular carcinoma) cell lines (Hep G2, SMMC-7721 and Bel-7402) in a dose-dependent as well as time-dependent manner, with IC_50_ values around 32 μM. Compared with human normal hepatocyte, HCC cell lines, especially HepG2 cells, were more sensitive in clavatustides-induced cell proliferation suppression. Their proliferation inhibitory property was due to suppression of G1/S phase transition [324]. Clavatustide B also exhibited cell growth inhibitory activity against PANC-1 and PC3, the chemo- and radio-therapy resistant cell lines. Cyclin E2 was found to be involved in clavatustide B-induced cell cycle arrest. With special origin and unique structure, these two compounds warrant further investigation to explore their underlying mechanisms and other potential bioactivities. Synthesis of clavatustides A and B analogs is also a meaningful approach to propel their future clinical application [325].

A novel cyclic tripeptide, sclerotiotide M (231) was isolated from the fungus *Aspergillus ochraceopetaliformis* DSW-2 and the compound only showed weak cytotoxic activities against human pancreatic cancer HPAC and BXPC3 cell lines with IC_50_ values over 20 μM [326].

Two cyclodepsipeptides, scopularides A (**232**) and B (**233**), were isolated from the fungus *Scopulariopsis brevicaulis*, which was found in the marine sponge *Tethya aurantium*. Both compounds significantly inhibited viability of several cancer cell lines including Colo357 (pancreatic adenosquamous carcinoma), Panc89 (pancreatic tumor) and HT29 cells at a concentration of 14.87 μM [327]. However, regorous and standard cytotoxic experimental procedures are needed to determine their anticancer potential.

The cytotoxic cyclodepsipeptide IB-01212 (**234**) was first isolated from the marine fungus *Clonostachys* sp. ESNA-A009. It was highly active against several cancer cell lines including LNCaP, SKBR3, HT29 and HeLa with the GI_50_ values on the order of 10^−8^ M [328]. In addition, IB-01212 also showed significant leishmanicidal activity through an apoptotic-like process, which added fresh insights to develop this compound as potential novel chemotherapeutic agent against *Leishmania* [329]. The synthetic and SAR studies of IB-01212 demonstrated great cytotoxic significance of its symmetric structure. The *N*-methylation on the amino acids of IB-01212 were found to enhance its cytotoxic effect, which were corroborating evidence for further discovery and synthesis of new analogs for anticancer drug development [330,331].

Microsporins A (**235**) and B (**236**) were cyclopeptides isolated from the marine fungus *Microsporum* cf. *gypseum* in bryozoan *Bugula sp.* from Virgin Islands in the US. Both were shown to be potent HDAC inhibitors and demonstrated cytotoxicity especially against HCT116 cell line, with IC_50_ values of 1.12 and 15.82 μM, respectively [332]. To extensively explore their potential bioactive properties, efforts towards the synthesis of microsporins and their isomers have been reported [333,334].

## 6. Conclusions and Future Perspectives

This review included the recently discovered natural anticancer cyclopeptides, which have been summarized and categorized according to their origins. An abundant number of cyclopeptides with diverging chemical and biological characteristics can be obtained from the aqueous and terrestrial portions of our nature. Cyclopeptides derived from natural sources have drawn great attention due to their high therapeutic potential for drug development. Today, many studies have shown that natural cyclopeptides are effective in the treatment and preventive measures of cancerous pathologies. Other biological activities such as antiviral, antibacterial and antimalarial effects have also been extensively observed whilst some are considered promising. Hence, detailed mechanism studies and structural modification of cyclopeptides are desperately needed for the development of cyclopeptide as therapeutic drug candidates. Nonetheless, the current research studies on the anticancer effects of natural cyclopeptides are just the tip of the iceberg. The natural cyclopeptides are undoubtedly a rich source as bioactive compound leads gifted from nature to benefit human welfare.

## Figures and Tables

**Figure 1 ijms-22-03973-f001:**
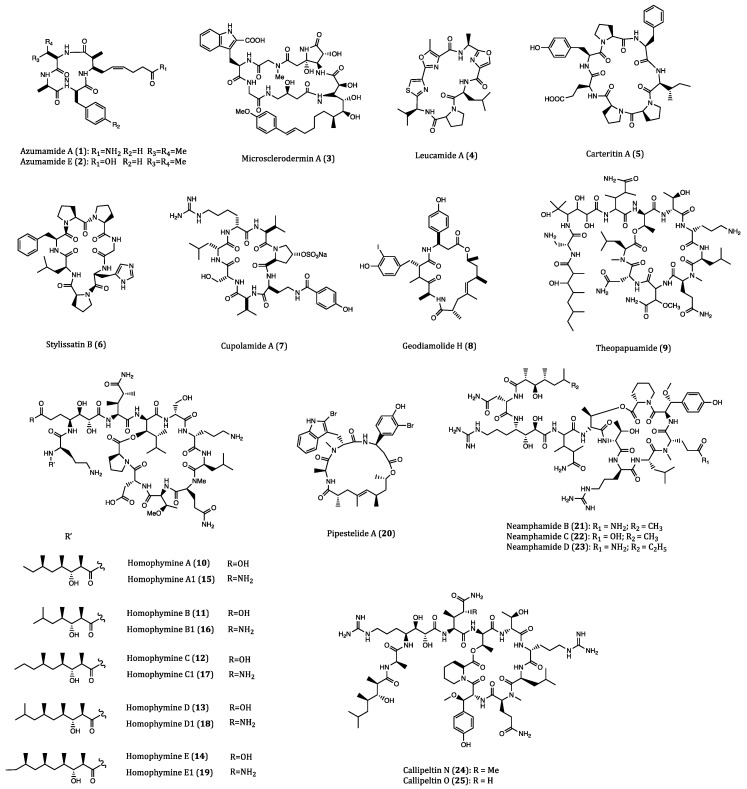
Structures of cyclopeptides **1**–**52** derived from sponges.

**Figure 2 ijms-22-03973-f002:**
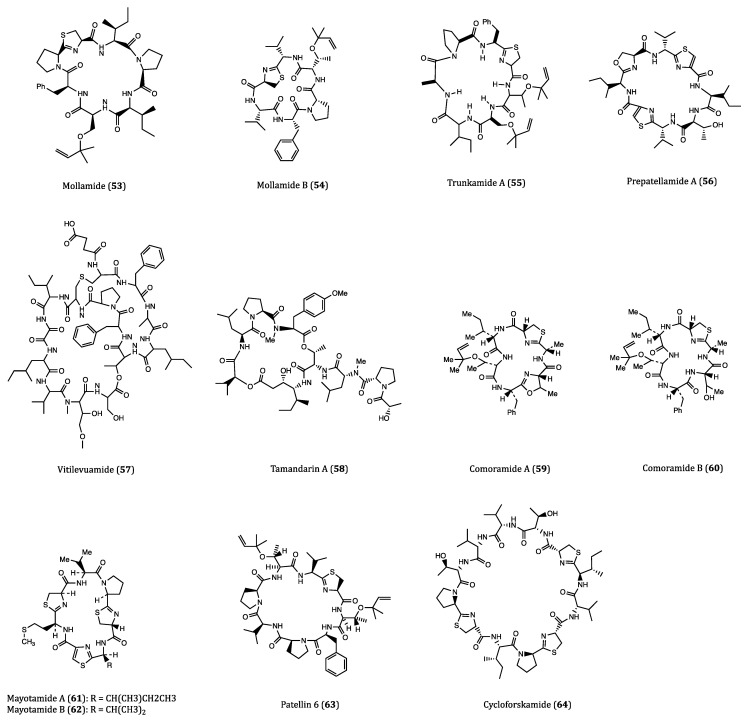
Structures of cyclopeptides (**53**–**64**) derived from ascidians/tunicates.

**Figure 3 ijms-22-03973-f003:**
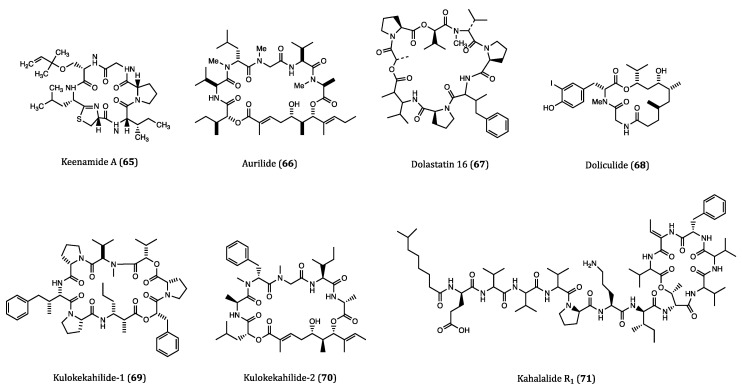
Structures of cyclopeptides (**65**–**71**) derived from ascidians/tunicates.

**Figure 4 ijms-22-03973-f004:**
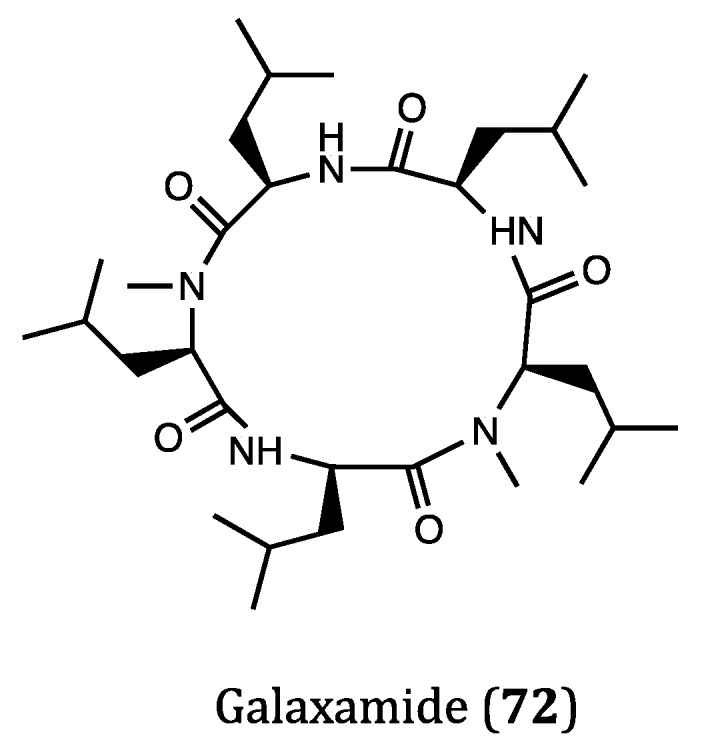
Structure of cyclopeptide (**72**) derived from marine algae.

**Figure 5 ijms-22-03973-f005:**
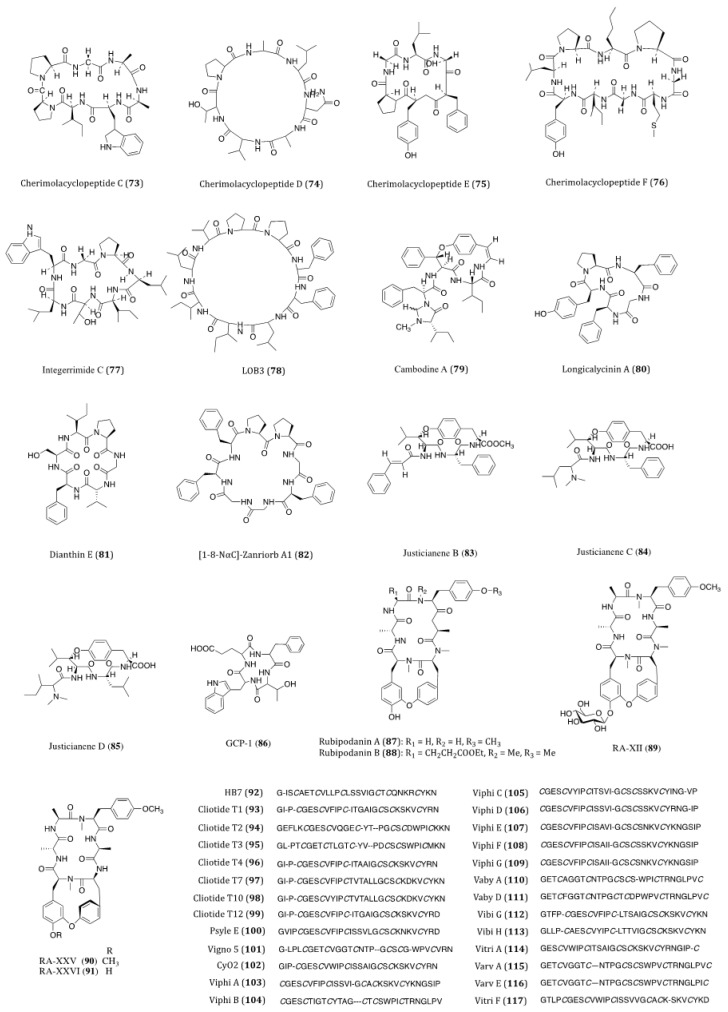
Structures of cyclopeptides (**73**–**117**) derived from terrestrial plant.

**Figure 6 ijms-22-03973-f006:**
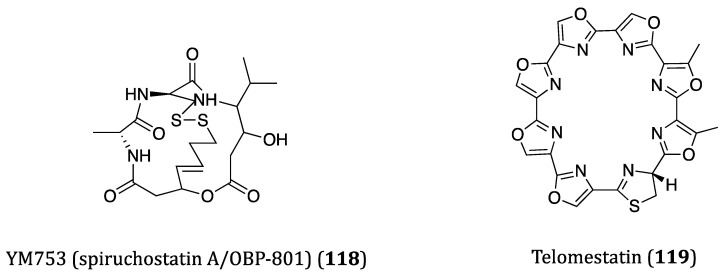
Structure of cyclopeptides (**118**–**119**) derived from non-marine bacteria.

**Figure 7 ijms-22-03973-f007:**
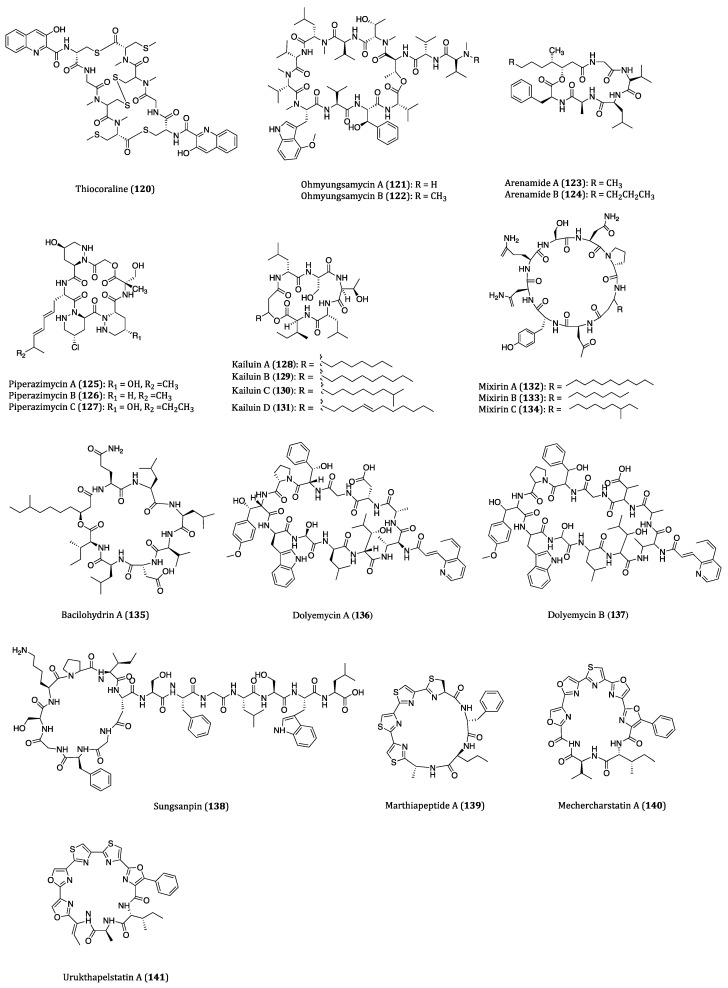
Structure of cyclopeptides (**120**–**141**) derived from marine bacteria.

**Figure 8 ijms-22-03973-f008:**
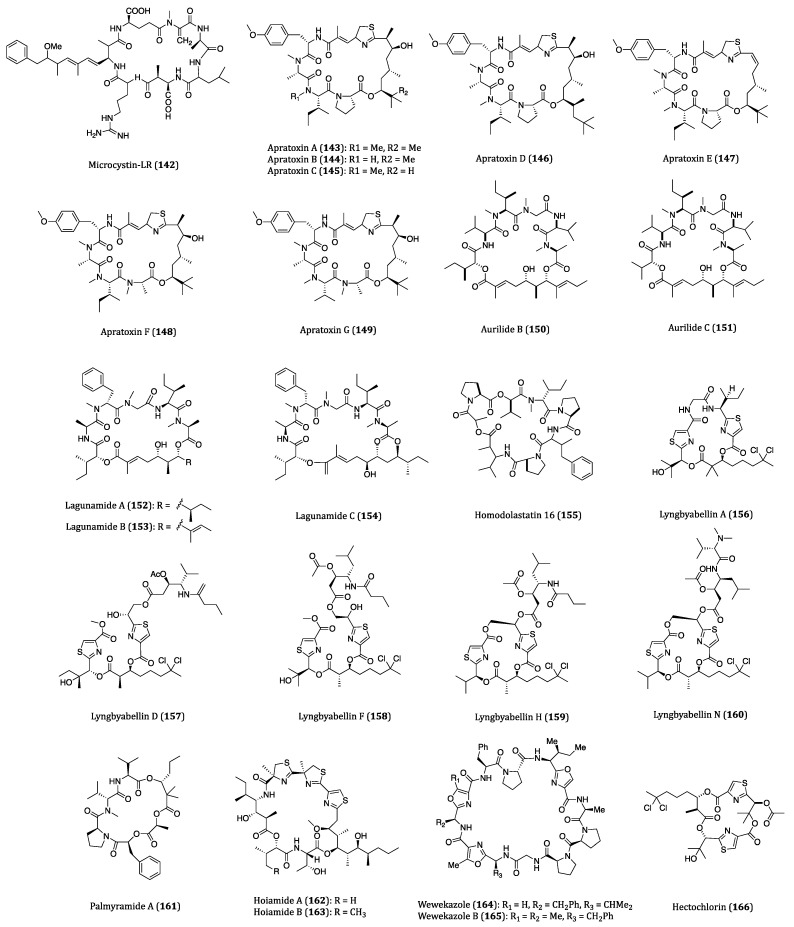
Structure of cyclopeptides (**142**–**204**) derived from cyanobacteria.

**Figure 9 ijms-22-03973-f009:**
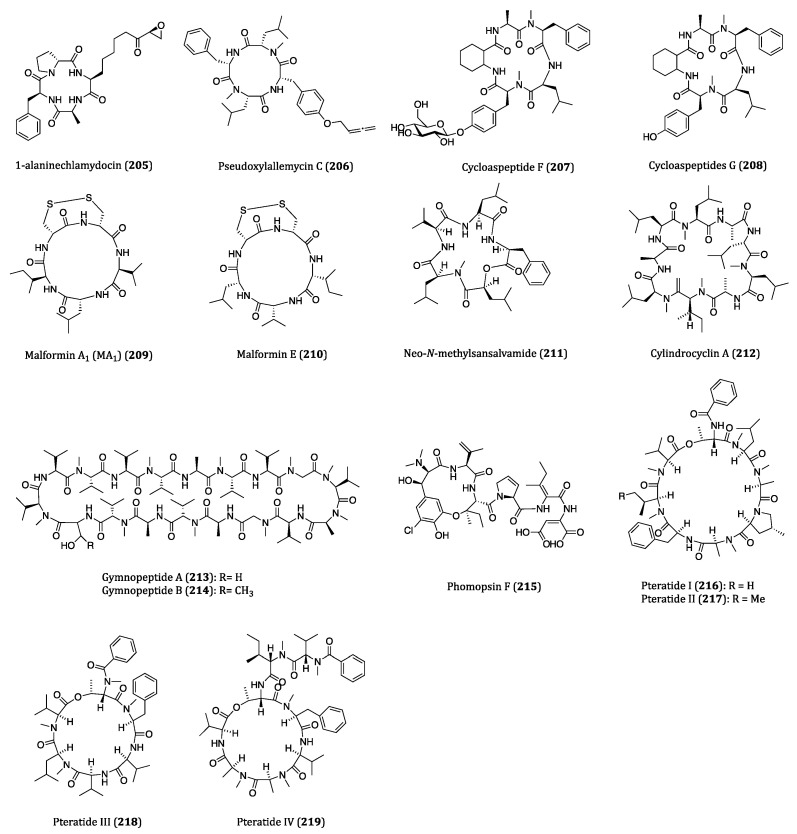
Structure of cyclopeptides (**205**–**219**) derived from non-marine fungi.

**Figure 10 ijms-22-03973-f010:**
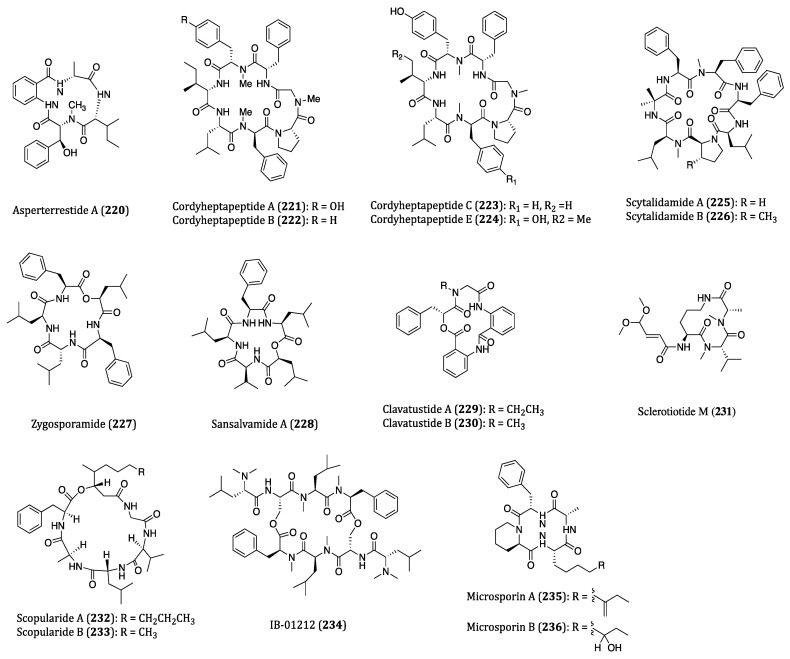
Structure of cyclopeptides (**220**–**236**) derived from marine fungi.

**Table 4 ijms-22-03973-t004:** Fungi-derived anticancer cyclopeptides.

Name	Biological Source	Anticancer Activity	Reference
Anticancer Cyclopeptides Derived from Non-Marine Fungi			
1-alaninechlamydocin (**205**)	*Tolypocladium* sp.	Cytotoxicity against MIA PaCa-2 (GI_50_ = 5.3 nM, LC_50_ = 22 nM) cell.	[296]
Pseudoxylallemycin C (**206**)	*Pseudoxylaria* sp. X802	Anti-proliferative effect to K562 (GI_50_ = 6.8 µM) cell. Cytotoxicity against HeLa (CC_50_ = 16.68 µM) cell.	[297]
Cycloaspeptides F (**207**) and G (**208**)	*Isaria farinosa*	Inhibitory activities toward MCF7 (GI_50_ = 18.7 and 15.2 µM, respectively) cell.	[298]
Malformin A_1_ (MA_1_) (**209**)	*Aspergillus niger*	Cytotoxicity against PC3 (IC_50_ = 130 nM), LNCaP (IC_50_ = 90 nM), HeLa (IC_50_ = 9.48 nM) and P388 (IC_50_ = 132.9 nM) cells.	[299,300,301]
Malformin E (**210**)	*Aspergillus tamarii*	Cytotoxicity against MCF7 (IC_50_ = 0.65 µM) and A549 (IC_50_ = 2.42 µM) cells.	[302]
Neo-*N*-methylsansalvamide (**211**)	*Fusarium solani* KCCM90040	Cytotoxicity against A549, SK-OV-3, SK-MEL-2 and MES-SA cells. (EC_50_ = 10.0 to 14.74 µM) Synergistic effect of neo-*N*-methylsansalvamide and paclitaxel when acting against MES-SA, HCT15, MES-SA/DX5 and HCT15/CL05 cells.	[303,304]
Cylindrocyclin A (**212**)	*Cyclindrocarpon* sp. strain A101-96	Cytotoxicity against COLO-320, HL-60, L1210 and human T lymphocyte Jurkat cells. (IC_50_~11 µM)	[305]
Gymnopeptides A (**213**) and B (**214**)	Mushroom, *Gymnopus fusipes*	Anti-proliferative activities against HeLa, A431, T47D, MCF7 and MDA-MB-231 cells. (IC_50_ = 18.0 to 88.4 nM and 14.0 to 44.3 nM, respectively)	[306,307]
Phomopsin F (**215**)	*Diaporthe toxica*	Impact on viability and vitability of HepG2 (IC_50_ = 6.1 µM) cell.	[308]
Pteratides I-IV (**216**–**219**)	*Pterula* sp.	Cytotoxicity against P388 (IC_50_ = 41 nM, 40 nM, 7.4 µM and 2.9 µM, respectively) cell.	[309]
Anticancer Cyclopeptides Derived from Marine Fungi			
Asperterrestide A (**220**)	*Aspergillus terreus* SCSGAF0162	Cytotoxicity against U937 (IC_50_ = 6.4 μM) and MOL-4 (IC_50_ = 6.2 μM) cells.	[310,311]
Cordyheptapeptide A (**221**)	*Cordyceps* sp. BCC 1788	Anti-proliferative activities to KB (IC_50_ = 0.78 μM), BC (IC_50_ = 0.2 μM), NCI-H187 (IC_50_ = 0.18 μM) and Vero (IC_50_ = 14 μM) cells.	[312,313]
Cordyheptapeptide B (**222**)	*Cordyceps* sp. BCC 16173	Anti-proliferative activities to KB (IC_50_ = 2.0 μM), BC (IC_50_ = 0.66 μM), NCI-H187 (IC_50_ = 3.1 μM) and Vero (IC_50_ = 1.6 μM) cells.	[312,313]
Cordyheptapeptide C (**223**)	*Acremonium persicinum*	Cytotoxicity against SF-268 (IC_50_ = 3.7 μM) and MCF7 (IC_50_ = 3.0 μM) cells.	[314]
Cordyheptapeptide E (**224**)	*Acremonium persicinum*	Cytotoxicity against SF-268 (IC_50_ = 3.2 μM), MCF7 (IC_50_ = 2.7 μM) and NCI-H460 (IC_50_ = 4.5 μM) cells.	[314]
Scytalidamides A (**225**) and B (**226**)	*Scytalidium* sp.	Cytotoxicity against HCT116 (IC_50_ = 2.7 and 11.0 µM, respectively) cell. Moderate cytotoxicitty in NCI 60 cell line panel. (Mean GI_50_ = 7.9 and 4.1 µM, respectively)	[315]
Zygosporamide (**227**)	*Zygosporium masonii*	Cytotoxicity against SF-268 (IC_50_ = 6.5 nM) and RXF393 (IC_50_ ≤ 5.0 nM) cells.	[316,317]
Sansalvamide A (**228**)	*Fusarium* sp.	Cytotoxicity against HCT116, COLO 205 and SK-MEL-2 (IC_50_ = 9.7, 3.46 and 5.84 μM, respectively) cells.	[318,319,320,321,322,323]
Clavatustide A (**229**)	*Aspergillus clavatus*	Anti-proliferative activity against Hep G2, SMMC-7721 and Bel-7402 cells. (IC_50_~32 μM)	[324]
Clavatustide B (**230**)	*Aspergillus clavatus*	Anti-proliferative activity against Hep G2, SMMC-7721 and Bel-7402 cells. (IC_50_~32 μM) Cell growth inhibitory activity against PANC-1 and PC3 cells.	[324,325]
Sclerotiotide M (**231**)	*Aspergillus ochraceopetaliformis* DSW-2	Cytotoxicity against HPAC and BXPC3 cells. (IC_50_ > 20 μM)	[326]
Scopularides A (**232**) and B (**233**)	*Scopulariopsis brevicaulis*	Viability inhibition of several cancer cell lines including Colo357, Panc89 and HT29 cells at a concentration of 14.87 μM.	[327]
IB-01212 (**234**)	*Clonostachys* sp. ESNA-A009	Inhibitory activity against several cancer cell lines including LNCaP, SKBR3, HT29 and HeLa with the GI_50_ values on the order of 10^−8^ M.	[328,329,330,331]
Microsporins A (**235**) and B (**236**)	*Microsporum* cf. *gypseum*	Cytotoxicity especially against HCT116 cell. (IC_50_ = 1.12 and 15.82 μM, respectively)	[332,333,334]

## Data Availability

Not applicable.

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
