# Peer review of "Natural Cyclopeptides as Anticancer Agents in the Last 20 Years"

_ijms, 2021, doi:10.3390/ijms22083973_

Round 1

Reviewer 1 Report

The submitted review is prepared based on 331 scientific articles. This number of quotable reports clearly indicated the importance of subject. I have found three recent reviews article concerning polypeptides and old one about plant cyclopeptides.

B. Pomilio, M. E. Battista, A. A. Vitale, Naturally-Occurring Cyclopeptides: Structures and Bioactivity, Current Organic Chemistry 10 (16) 2006; 2075-2121; N. H. Tan, J. Zhou, Plant cyclopeptides, Chem. Rev., 106 2006 840-895. P. Wipf, Synthetic studies of biologically active marine cyclopeptides, Chem. Rev., 95 1995, 2115-2134.

The body text is divided into parts according to the habitat of microorganisms producing particular cyclopeptides. It is logical arrangement which makes easily movement in text. The examples of leading structures are concentrated in figures and the activity is presented in transparent forms of tables.

From the formal point of view discussed cyclopeptides are oligopeptides with number of amino acids 4-8 with some exceptions where the number of units in larger. The term cyclopeptides is rather a common name, and it should be mentioned on the beginning of introduction. Inn text is also mentioned Telomestatin (119) which is a macrocyclic compound but not cyclopeptide and this compound (page 29) should be remove. In general, all abbreviation for cell lines should be explained when first time were used or listed on the beginning of the body text. The references have to be completed by full bibliographic data, volume, year, pages; e.g. position 12, 13 is not completed.

Authors should make carefully inspection of text to remove type writing errors.

Abstract: Cyclopeptides or cyclic peptides are referred to polypeptides possessing a cyclic ring structure of amino acids? I think should be better: Cyclopeptides or cyclic peptides are polypeptides formed by ring closing of terminal amino acids.

Page 3 l. 108: IC50 values of 0.045 unit is lost

Page 4 l. 144: lambda capital (λ) have to be change by gamma (g)

Page 4 l.151: deleteriouseffect change to deleterious effect

Page 5 l.234: it is really 41.8 mM?

Page6 l.298: comprehensivechemical change to comprehensive chemical

Page 11 l. 373 In addition, The compound change to in addition, the compound

Page 14 l. 484; Page 37 l.113: are desperately needed? may be: are urgently needed

Page 14 l.503 phasewas change to phase was

Page 21 l. 613 was nokt change to was not

Page 23 l. 694 at altitudes over change to at attitudes over

Page 28 l. 727 quite a number of; may be: numerous of

Page 30 l. 801 what means: a nitrogen-containing methyl group, please explain

Page 30 l. 809 and 822: analogs or analogues please use one form

Page 37 l. 115: xenografts was largely reduced; could you please specified in percentage or millimetres.

Page49 l.129 versatil change to versatile

Page 52 l. 138: Behamas change to Bahamas

Page 53 l. 144 Modification of its methyl groups? What it means?

So, I recommend this paper for publication after several minor improvements listed above.

Author Response

Please see our replies in the attached file. Thanks.

Reviewer 2 Report

Natural Cyclopeptides As Anticancer Agents in Recent 20 Years

Jia-Nan Z hang, Yi-Xuan Xia and Hong-Jie Z hang

The review “Natural Cyclopeptides As Anticancer Agents in Recent 20 Years” deals with the anticancer cyclopeptides discovered in last twenty years.

The authors highlight the advantages related to the discovery of new bioactive cyclic peptides to be used as potential candidates in the anticancer drug discovery. Structural rigidity, biochemical stability, binding affinity and membrane permeability are the main advantages of cyclopeptides compared not only to linear peptides but also to other chemical scaffolds.

The review is basically well written and the cyclopeptides are classified on the basis of their origin. The good organization of the arguments makes the review easy to read. The authors also deal with SAR studies and the total synthesis of the compounds. These points make the work complete, since it offers a general overview of anticancer cyclopeptides of natural origins.

Recommendations:                 Accepted after minor revision

Some minor issues have to be addressed.

Line 326: “Nd-carbamoyl Asn”, please correct in “Nd-carbamoyl Asn”

Line 805: please remove the additional space.

Line 1034, 1058: please change “L. majuscule” in “L. majuscule

Line 1277-1283: please modify the font in accordance with the rest of the manuscript.

Author Response

(The authors gave the same response as above.)
